# Interpretable multimodal classification for age-related macular degeneration diagnosis

Carla Vairetti [1,4]*, Sebastián Maldonado[2,4], Loreto Cuitino[3,5], Cristhian A. Urzua[3,6]*

**1** Facultad de Ingeniería y Ciencias Aplicadas, Santiago, Chile, **2** Department of Management Control and Information Systems, School of Economics and Business, University of Chile, Santiago, Chile, **3** Laboratory of Ocular and Systemic Autoimmune Diseases, Faculty of Medicine, University of Chile, Santiago, Chile, **4** Instituto Sistemas Complejos de Ingeniería (ISCI), Santiago, Chile, **5** Servicio de Oftalmología, Hospital Clínico Universidad de Chile, Santiago, Chile, **6** Faculty of Medicine, Clinica Alemana-Universidad del Desarrollo, Santiago, Chile

* cvairetti@uandes.cl (CV); cristhian.urzua@uchile.cl (CAU)

## Abstract

Explainable Artificial Intelligence (XAI) is an emerging machine learning field that has been successful in medical image analysis. Interpretable approaches are able to "unbox" the black-box decisions made by AI systems, aiding medical doctors to justify their diagnostics better. In this paper, we analyze the performance of three different XAI strategies for medical image analysis in ophthalmology. We consider a multimodal deep learning model that combines optical coherence tomography (OCT) and infrared reflectance (IR) imaging for the diagnosis of age-related macular degeneration (AMD). The classification model is able to achieve an accuracy of 0.94, performing better than other unimodal alternatives. We analyze the XAI methods in terms of their ability to identify retinal damage and ease of interpretation, concluding that grad-CAM and guided grad-CAM can be combined to have both a coarse visual justification and a fine-grained analysis of the retinal layers. We provide important insights and recommendations for practitioners on how to design automated and explainable screening tests based on the combination of two image sources.

## Introduction

In the clinical management of early-stage age-related macular degeneration (AMD), identifying patients at high risk of progressing to vision-threatening complications is both crucial and challenging [1]. With an aging global population and a shortage of specialists, particularly in developing countries, there is an urgent need for tools that facilitate rapid patient screening [2].

Automated deep learning (DL) systems have demonstrated performance on par with ophthalmologists [3]. Ophthalmic diagnosis is inherently multimodal [4, 5]. Optical coherence tomography (OCT) provides cross-sectional images of the retina, revealing its distinct layers, while fundus imaging offers a two-dimensional (2D) view of the three-dimensional (3D) retinal tissues. Various fundus imaging techniques, such as infrared reflectance (IR), near-infrared reflectance (NIR), fundus autofluorescence (FAF), and color fundus photographs (CFP), have been explored in AI applications [6–8].

**Data Availability Statement:** Our data cannot be made publicly available for ethical and legal reasons. Public availability would compromise patient privacy, and public deposition would breach compliance with the protocol approved by our

research ethics board (the original files and their translations were attached in the previous submission). The data associated with this study can be shared privately upon reasonable request. Please direct any data access requests to Dr. Fabian Vega Tapia at the Laboratory of Ocular and Systemic Autoimmune Diseases, Faculty of Medicine, Universidad de Chile. Dr. Vega Tapia can be reached via email at fabian.vega@uchile.cl.

**Funding:** The authors gratefully acknowledge financial support from ANID PIA/PUENTE AFB230002 and FONDECYT-Chile, grants 12200007, 1200221, 11191215, and 1212038. The funders had no role in study design, data collection and analysis, decision to publish, or preparation of the manuscript.

**Competing interests:** The authors have declared that no competing interests exist.

This paper introduces a novel DL-based system for multimodal AMD diagnosis, incorporating Explainable Artificial Intelligence (XAI) strategies to make AI systems more interpretable in medical image analysis. The framework not only provides accurate diagnoses but also allows practitioners to visually inspect and understand the AI's decision-making process. We evaluate three XAI techniques—Grad-CAM, guided backpropagation, and guided Grad-CAM—comparing their diagnostic effectiveness and interpretability.

The study makes two key contributions:

- It introduces the first interpretable AI system that integrates both IR and OCT imaging for multimodal AMD diagnosis, addressing the challenge of explainability in this domain. Previous XAI studies in ophthalmology have focused on a single imaging source [9], while existing multimodal OCT systems have not emphasized XAI techniques [5, 10].

- This study presents a novel AI system for AMD diagnosis, implemented at a major hospital in Chile. While research on AI in ophthalmology is growing [10], clinical implementation has been hindered by the challenges of explainability and interpretability. This work addresses these gaps by providing insights into the methodologies used by XAI approaches to categorize ocular images, marking a significant step towards the clinical translation of AI in ophthalmology. Experimental results from this proprietary dataset demonstrate the system's practical value.

The remainder of this paper is organized as follows: First, we review XAI approaches for medical diagnosis. Next, we describe the proposed multimodal framework for interpretable AMD diagnosis. We then report experimental results using various DL and XAI approaches. Finally, we conclude with recommendations for practitioners and discuss future developments.

## Theoretical background on multimodal OCT imaging and XAI techniques

Deep learning has significantly advanced ophthalmology, particularly in the early detection of AMD through imaging analysis [10]. CNN models have replaced the traditional two-step process of feature extraction and machine learning, outperforming other techniques and matching the performance of experienced ophthalmologists [3, 11].

Most CNN-based studies on AMD detection rely on a single data source, such as OCT or fundus imaging, with popular architectures including MobileNet, ResNet, AlexNet, GoogleNet, VGG, and Xception [11, 12]. Beyond AMD detection, related conditions like diabetic macular edema (DME) and choroidal neovascularization (CNV) are also analyzed [11–13].

Some studies have explored multimodal approaches to AMD detection. Notably, Vaghefi et al. [14] combined OCT and CFP imaging using Inception-ResNet-v2, while Yoo et al. [15] used restricted Boltzmann machines and deep belief networks for a similar combination. Although both studies show that combining different data sources is more effective than using a single image, their small sample sizes ($n = 75$ and $n = 83$, respectively) are limitations. A larger study ($n = 2450$) by Keenan et al. [3] combined FAF and CFP to detect reticular pseudodrusen (RPD), though it excluded OCT imaging.

Recent research has shifted towards analyzing various imaging modalities and AMD variants, such as dry AMD [8]. Wang et al. [16] employed a multimodal approach using OCT and CFP imaging, while Anegondi et al. [6] used multimodal deep learning to predict geographic atrophy (GA) growth rates with OCT and FAF imaging. Similarly, Winkler et al. [2] analyzed GA using FAF and fundus photos (FP), and Oncel et al. [5] studied the relationship between intraretinal hyperreflective foci (IHRF) on OCT B-scans and hyperpigmentation on CFP. Goh

et al. [1] demonstrated the benefits of multimodal imaging over CFP alone by combining OCT, FAF, NIR, and CFP.

Explainability is a key focus in this study. Most XAI studies in medical imaging use a single data source, with X-ray, MRI, and CT scans being the most common [17, 18]. Ophthalmic studies predominantly focus on the chest and brain, followed by the eye. Van der Velden et al. [18] reported 19 studies using fundus images and five using OCT.

Regarding XAI techniques in eye imaging, Class Activation Mapping (CAM) and Grad-CAM are the most prevalent [18]. CAM, one of the earliest and most widely-used CNN visualization methods, produces a saliency map to show the contribution of input features via a heatmap [19]. Variants like Guided Grad-CAM [20] and multiple instance learning [21] are also used.

For AMD diagnosis, several studies have applied XAI strategies to either fundus or OCT imaging [9, 18]. Most focus on diabetic retinopathy [21, 22] or glaucoma [23], with similar tasks applied to OCT imaging [24, 25]. To our knowledge, only Perdomo et al. [24] and Wang et al. [16] have used XAI approaches to detect AMD.

Aside from Wang et al. [16], where CAM is used for pattern visualization, no prior studies have introduced a multimodal XAI framework for AMD detection. In our research, we compare three modern XAI techniques (Grad-CAM, Guided Backpropagation, and Guided Grad-CAM). This analysis is important because, while CAM offers a coarse heatmap, Guided Backpropagation and Guided Grad-CAM provide detailed visualizations of retinal layers. A key differentiator of our study is the use of IR imaging instead of other types of fundus images.

A few XAI studies have explored multimodal learning for medical diagnosis [4]. For example, Lee et al. [26] developed a novel DL approach that explains diagnostic decisions with both heatmaps and textual justifications, though it is based on a single image. Holzinger [27] proposed a path toward multimodal causability using XAI methods, but did not present a concrete DL framework for achieving this.

## Proposed framework for AMD detection using multimodal XAI

In this section, we propose a framework for AMD detection based on the combination of IR and OCT imaging. After image pre-processing, we consider a standard two-step strategy, in which we first train a multimodal deep learning architecture and subsequently apply several XAI strategies. This study discusses the following research questions:

**RQ1:** Is infrared reflectance a valuable data source for AMD prediction, in the sense that its combination with OCT imaging leads to a better predictive performance in relation to OCT alone?

**RQ2:** Does the generation of visual justifications for both IR and OCT imaging leads to a better interpretation for AMD diagnosis?

**RQ3:** Does the combination of two or more XAI methods provides a better interpretation for AMD diagnosis in relation to a single strategy?

Notice that this approach can be extended easily to other fundus images, such as NIR, FAF, and CFP. Furthermore, the approach can be used for detecting other entities that affect the macula, including DME, hydroxychloroquine maculopathy, macular dystrophies, among others. Finally, the classification task can be adapted from binary to multiclass learning in order to detect different retinal conditions with the same model. Multiclass approaches for detecting retinal diseases have been widely discussed in the literature (see, e.g., [11, 24]).

The multimodal architecture is described next. For each patient, the input data corresponds to a single block containing both the IR and OCT images. We first process this image by cropping and resizing them to train them independently. We explore the performance of different CNN architectures, which are described next. The two DL models are combined with a single prediction layer using the late fusion technique.

The following CNN architectures are studied in this paper (see [19] for a detailed description):

- *Inception-v3* is a 48-layers deep CNN from the Inception family. It includes several improvements in relation to the original version, such as factorized 7 x 7 convolutions and label smoothing.

- *Xception* is a 71-layers deep CNN that extends the Inception architecture. It replaces the Inception modules with depthwise separable convolutions.

- *ResNet-50* is a popular variant of the ResNet model that considers 48 convolution layers along with one MaxPool and one average pool layer.

- *MobileNet-v2* is 53 layers-deep model that considers depth-wise separable convolutions to construct lightweight CNNs. The model has an inverted residual structure in which non-linearities associated to thin layers are removed. MobileNet-v2 is an updated version of the original model with higher efficiency, being almost twice as fast as MobileNet-v1.

The four architectures considered in this study —ResNet, Xception, MobileNet, and Inception— were selected for consistency with recent AMD detection research. For example, Tasnim et al. [11] found MobileNet to perform best, while Vaghefi et al. [14] achieved strong results using Inception and ResNet in a multimodal framework. Although other architectures like AlexNet, VGG, and DenseNet have been used in combination with these models in past studies [3, 12, 13], they no longer represent the state of the art.

Transfer learning plays a crucial role in the success of CNNs, particularly in medical diagnosis where the number of available images is often limited. By adapting pre-trained models from large datasets, the need for extensive data is reduced [28]. In this study, we utilized pre-trained CNN classifiers, initially trained on the widely-used ImageNet database, which contains over a million labeled real-world images. These classifiers were then fine-tuned using our dataset.

Regarding the performance of XAI methods, the following strategies are discussed in this paper [29]:

- *Grad-CAM* represents an evolution of the CAM approach. Grad-CAM takes the gradients of the target object into account to produce a coarse heat-map, emphasizing on the regions of the image that are relevant to distinguish the concept (retinal damage in our case) from other objects. The use of gradients confers versatility and accuracy to the approach [29]. Formally, Grad-CAM computes the gradients of $y^c$, the score for class $c \in \{Normal, AMD\}$, with respect to the activations $A_k$ of a convolutional layer, for a given feature map $k$. The scores are the outputs of the model that result from applying the decision function to each data point, and can be cast into probabilities using the softmax function.
  The computation of the gradients is done before applying the softmax function. The importance weights $\alpha_c^k$ are obtained by aggregating the gradients over the width and height ($i$ and $j$, respectively) of the feature map via global average pooling:

$$\alpha_c^k = \frac{1}{Z} \sum_i \sum_j \frac{\partial y^c}{\partial A_{ij}^k} \qquad (1)$$

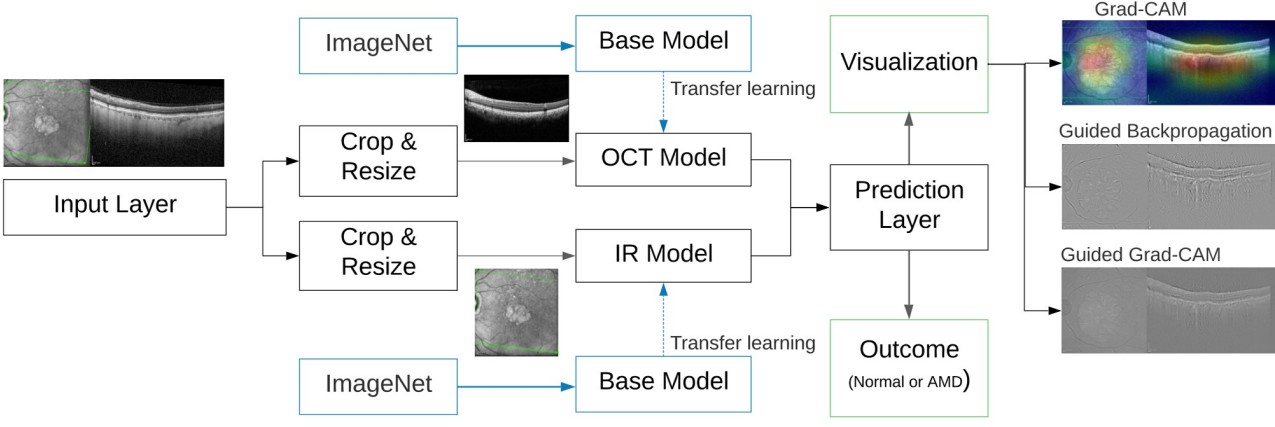

**Fig 1. Diagram for multimodal AMD prediction via CNNs and XAI techniques.**

The localization map $L^c$ computed for class $c$ is obtained by performing a linear combination of the $\alpha_c^k$ values and the activations, and the applying the Rectified Linear Unit (ReLU) function [29]:

$$L^k = ReLU(\sum_k \alpha_c^k A^k) \tag{2}$$

- *Guided backpropagation* makes modifications to the raw gradients to provide a visual explanation of a CNN. In contrast to Grad-CAM, it is not class-discriminative in the sense that it generates almost similar visualizations for a given image regardless of the target concept [29]. The output of the model highlights the contour of the objects instead of providing a heat-map, resulting in an interesting alternative to Grad-CAM. This is done by visualizing the positive gradients with respect to the image, suppressing the negative ones when back-propagating to the activation layers.

- *Guided Grad-CAM* combines the two previous methods, capturing the best of the two worlds: it is a class-discriminative approach as it is able to locate relevant regions of the target concept, while generating a fine-grained output that defines the contours of it. This is achieved by fusing the two visualizations using element-wise multiplication [29].

The proposed framework is summarized in Fig 1. After cropping and resizing the input image, the corresponding regions are processed by two CNNs, the OCT and IR models. In both networks, we consider a transfer learning approach based on ImageNet. The prediction layer performs the late fusion, combining the two networks to provide a single outcome (either normal or AMD). The output of the proposed framework is twofold: first, the diagnosis is generated for a new image, while a visual justification is also presented by considering three XAI techniques. Based on our results, we propose using two XAI strategies only: Grad-CAM and Guided Grad-CAM. This is discussed in the last section of the manuscript.

Finally, the evaluation of the performance of the various XAI methods discussed in this paper is more subjective than of the performance of the classifiers, and rely on the opinions of the team of ophthalmologists in the project.

Before applying the XAI techniques, we select the best classification strategy using the Area Under the ROC Curve (AUC-ROC) as performance metric on a test set, which remains

unseen during the model calibration process. This decision includes the unimodal or multimodal strategies and the CNN architecture.

The AUC-ROC represents the area under the receiver operating characteristic (ROC) curve, which plots the true positive rate (TPR) against the false positive rate (FPR) across all possible threshold settings in the training set. Unlike metrics such as accuracy or F1, AUC does not necessitate the selection of a classification threshold. Thus, the best model can be identified using this metric, and subsequently, the threshold can be adjusted based on desired TPR and FPR values. This metric also ensures a balanced accuracy between classes.

When choosing among XAI techniques, we consider criteria such as the ability of the final users to interpret the visual justification provided by the method in a straightforward manner (easy-to-use visual tool). Furthermore, we assess the ability of the model to highlight retinal damage in a clear manner in the cases that have this condition.

## Experimental results

We applied the three XAI methods discussed in the previous section to a dataset collected by the team of ophthalmologists in the project. First, a description of this dataset is presented, including also the experimental settings. Next, the performance of the different CNN classifiers and unimodal/multimodal approaches is summarized. Finally, a discussion of the performance of the various XAI classifiers is provided for specific instances in the test set.

### The dataset and experimental setting

We followed a four-step framework for the experimental design, which consists in (1) data collection, (2) labeling and exclusion criteria, (3) model training and evaluation, and (4) visual justifications.

We collected 4563 images of patients seen at the Hospital Clínico Universidad de Chile in Santiago, Chile. This dataset was obtained by scholars and practitioners of the Department of Ophthalmology of the University of Chile. Each record consists in the combination of an IR and an OCT image extracted from across the macula cube area ($8600 \, \mu m \times 7167 \mu m$ or $30\degree \times 25\degree$). The device used to acquire the images corresponds to a Spectralis OCT 2 imaging platform (Heidelberg Engineering, Germany).

Each scan was manually labeled as 'AMD' (dry or wet AMD), 'NORMAL,' 'OTHER,' or 'UNDEFINED' by two experienced retinal specialists. Discrepancies between specialists led to a label of 'UNDEFINED,' which, along with images containing diagnostic artifacts or with a signal strength index (SSI) lower than 7, were excluded from the analysis. Artifacts were identified following established criteria [30, 31]. The 'OTHER' category included findings indicative of conditions other than AMD, such as vitreo-macular traction, macular holes, and retinal/RPE detachments [32].

The category 'OTHER' represents findings that may be found in diseases other than AMD [32], such as: alterations related to vitreo-retinal interface (vitreo-macular traction, lamellar and full-thickness macular hole, epiretinal membrane), suggestive signs of macular edema (intraretinal hyporeflective spaces) or vascular exudation (intraretinal hipereflective dots), retinal/RPE detachments and alterations in normal macular structure (pathologic myopia).

Records labeled as 'OTHER' or 'UNDEFINED' were discarded, totaling 237, resulting in a final dataset comprised of 4326 images. These exclusions enable the machine to learn from a more refined dataset, thereby reducing model noise. It is important to note that OCT/Fundus images were selected from patients with a potential AMD diagnosis and do not represent all patients with retinal pathologies treated at the Hospital. Images from healthy controls were individually selected to match the age and sex of each of the patients with potential AMD.

However, this process was carried out before applying the exclusion criteria, leading to a greater number of normal cases than AMD samples in the final dataset.

Of the 4326 images, 1948 were AMD cases, including alterations like CNV, drusen, and subretinal fluid. The remaining 2378 images were from healthy patients. Images from both eyes were included if the patient had possible AMD findings in both eyes. Data was collected retrospectively from medical records between June 1, 2019, and June 30, 2021. Participant information was anonymized, with only the researchers involved in data collection having access to identifiable details.

The model evaluation was performed as follows: a nested validation strategy was made, splitting the data into a training set (70% of the samples) and a test set (30% of the samples). Ten-fold cross-validation was performed in the training set in order to define the number of epochs and to obtain an average performance. For each set of experiments given by the different combination of data sources (OCT/IR only and the combinations), the models were trained for an increasing number of epochs (from 1 to 15). The best validation performance, computed as the average for all the CNN classifiers, provides the optimal number of epochs for a given set of experiments. Notice that all classifiers have the same number of epochs within a set of experiments.

In terms of the model parameters, we considered a batch size of 16, with images of size $248 \times 632$ pixels. We used the well-known Adam optimizer, with initial learning rates of 0.0001 and 0.00001 for pre-training and fine-tuning, respectively, and $\beta_1 = 0.9$ and $\beta_2 = 0.999$. The default configurations were considered for the additional parameters of each CNN classifier. As a standard binary classification problem, cross-entropy was considered as the loss function, and the Rectified Linear Unit (ReLU) as the activation function.

In order to make the training process more efficient, we used a 'warm start' procedure, in which the weights obtained in a previous iteration are used as a starting point for the network in the following iteration [33]. All the experiments were performed in Python using the scikit-learn and Keras libraries. The codes can be obtained from the authors upon request. Unfortunately, the dataset cannot be shared due to the rules of confidentiality regarding patient information.

The flowchart of this process is presented in Fig 2, including the visualization step in which Grad-CAM and Guided Grad-CAM are considered.

## Classification performance summary

The performance of the four CNN architectures discussed in the previous section (Inception, ResNet, MobileNet, and Xception) is evaluated with our dataset considering four different unimodal/multimodal strategies to deal with the two images: OCT alone (the IR image is discarded), IR alone (the OCT image is discarded), OCT and IR combined as a single image, and the proposed multimodal strategy based on late fusion. Notice that the third strategy is unimodal since there is a single image as input.

The performance of the various unimodal and multimodal strategies and CNN classifiers is summarized in Table 1. For each approach, the average AUC-ROC/accuracy of the cross-validation strategy and the test performance for the holdout step are reported. The best test performance is highlighted in bold.

Table 1 shows that the multimodal architecture combining IR and OCT imaging achieves the best performance across both metrics, with the highest AUC for all classifiers. This result highlights the benefit of incorporating IR imaging to complement OCT scans and demonstrates the effectiveness of a late fusion multimodal strategy. We also observe that AUC and accuracy are relatively similar when OCT and IR imaging are used independently.

| Data collection | Labeling and exclusion criteria | Training and Evaluation | Visualization |
|---|---|---|---|
| We collected 4563 images of patients seen at the Hospital Clínico Universidad de Chile in Santiago, Chile.<br><br>This dataset was obtained by scholars and practitioners of the Department of Ophthalmology of the University of Chile.<br><br>Each record consists in the combination of an IR and an OCT image extracted from across the macula cube area using a Spectralis OCT 2 imaging platform. | Each scan was manually labeled as 'AMD' (dry AMD or wet AMD), 'NORMAL', 'OTHER', 'UNDEFINED' by two experienced retinal specialists. If the annotations did not coincide, the images were categorized as 'UNDEFINED'. Images that presented alterations that may prevent the correct diagnosis were also labeled as 'UNDEFINED'. The category `OTHER` represents findings that may be found in diseases other than AMD. Records labeled as `OTHER` or `UNDEFINED` were discarded. | The model evaluation was performed as follows: a nested validation strategy was made, splitting the data into a training set (70% of the samples) and a test set (30% of the samples). Ten-fold cross-validation was performed in the training set in order to define the number of epochs and to obtain an average performance (AUC as the performance metric). For each set of experiments (OCT/IR only and the combinations), the models (Inception, ResNet, MobileNet, and Xception) were trained until 15 epochs. | We consider Grad-CAM and Guided Grad-CAM as XAI methods. The first one provides a coarse visual justification, highlighting the regions in which the retinal damage can be located. The Guided Grad-CAM, in contrast, provides a fine-grained visualization of the different layers of the retina.<br><br>This two-step analysis resembles the procedure followed by an ophthalmologist to reach a diagnosis: it first examines the area of interest and then inspect the profiles of the retina. |

**Fig 2. Flowchart of the four-step framework followed in the experimental section.**

The validation performance from cross-validation closely matches the test performance on the unseen 30% of samples, suggesting a low risk of overfitting—a crucial factor given the smaller datasets typical in medical tasks compared to other computer vision applications. Overfitting was further monitored by analyzing the loss function across increasing epochs.

To determine statistical significance, we applied the Friedman test with the Iman-Davenport correction, as well as the Holm test, both of which are widely used for comparing different machine learning methods [34]. The Friedman test computes a chi-squared statistic to assess

**Table 1. Validation/test performance for the various CNN classifiers and unimodal/multimodal strategies.** AUC-ROC and accuracy as evaluation measures. The standard deviation of the validation performance is presented in parenthesis.

| CNN | OCT only | IR only | uni. OCT+IR | multi. OCT+IR |
|---|---|---|---|---|
| *AUC as performance metric* | | | | |
| Inception | 0.972 (0.004)/ 0.955 | 0.979 (0.004)/ 0.970 | 0.970 (0.005)/ 0.965 | 0.970 (0.004)/ 0.974 |
| ResNet | 0.973 (0.005)/ 0.965 | 0.971 (0.006)/ 0.971 | 0.966 (0.007)/ 0.969 | **0.977 (0.004)/ 0.979** |
| Xception | 0.957 (0.008)/ 0.958 | 0.969 (0.007)/ 0.967 | 0.959 (0.007)/ 0.960 | 0.972 (0.005)/ 0.974 |
| MobileNet | 0.962 (0.006)/ 0.971 | 0.968 (0.008)/ 0.968 | 0.971 (0.005)/ 0.970 | 0.979 (0.003)/ 0.978 |
| *Accuracy as performance metric* | | | | |
| Inception | 0.928 (0.012)/ 0.910 | 0.927 (0.013)/ 0.919 | 0.925 (0.010)/ 0.920 | 0.934 (0.012)/ 0.915 |
| ResNet | 0.928 (0.013)/ 0.915 | 0.926 (0.014)/ 0.918 | 0.929 (0.013)/ 0.925 | **0.937 (0.013)/ 0.940** |
| Xception | 0.917 (0.015)/ 0.925 | 0.924 (0.011)/ 0.907 | 0.913 (0.015)/ 0.925 | 0.943 (0.017)/ 0.935 |
| MobileNet | 0.935 (0.011)/ 0.930 | 0.929 (0.015)/ 0.921 | 0.917 (0.016)/ 0.925 | 0.933 (0.011)/ 0.925 |

**Table 2. Average results and Holm test for pairwise comparisons for the four approaches (OCT only, IR only, uni. OCT+IR, and multi. OCT+IR) and the two performance metrics (AUC-ROC an accuracy).**

| Method | AUC-ROC | Accuracy | Rank | p-value | $\frac{\alpha_Z}{(k-1)}$ | Outcome |
|---|---|---|---|---|---|---|
| multi. OCT+IR | 0.976 | 0.929 | 1.438 | - | - | - |
| uni. OCT+IR | 0.966 | 0.924 | 2.500 | 0.100 | 0.050 | not reject |
| IR only | 0.969 | 0.916 | 2.875 | 0.026 | 0.025 | not reject |
| OCT only | 0.962 | 0.920 | 3.188 | 0.007 | 0.017 | reject |

whether significant differences exist among the four approaches (OCT only, IR only, uni. OCT +IR, and multi. OCT+IR). The test yielded a value of 3.893, with a p-value below 0.023, allowing us to reject the null hypothesis of similar performances.

The Holm test conducts pairwise comparisons between the top-performing approach and the others using Nemenyi's Z test [34]. Table 2 presents the p-value of each test, the significance threshold $\alpha_Z/(k-1)$ (with $k = 2, 3, 4$ and $\alpha_Z = 0.05$), and the test outcomes in the fifth, sixth, and seventh columns, respectively. A 'reject' outcome is indicated when the p-value falls below the corresponding threshold.

In Table 2, we observe that the multimodal approach combining OCT and IR imaging achieves the best average performance across various CNN classifiers, as indicated by the highest AUC-ROC and accuracy values (third and fourth columns, respectively) and the best overall ranking (lowest value in the fifth column). Although this approach does not statistically outperform all the remaining methods (it only outperforms OCT as a single source), we conclude that an average rank of 1.438 across four techniques is a promising result for the proposed framework.

## Results for the XAI methods and discussion

In the following figures, we discuss the visual justifications for the best classifier, which is the proposed multimodal strategy with the OCT and IR images as inputs, and ResNet as CNN classifier. Each figure presents four images: the input image, and the visualization of the three XAI methods: Grad-CAM, Guided Backpropagation, and Guided Grad-CAM. his analysis was designed and validated by Prof. C. Urzua, an experienced ophthalmologist and co-author of the paper. He also defined the ground truth (represented by red squares in the figures) and the regions of interest in the images.

Next, we analyze six different illustrative examples from the test set to discuss the usefulness of the visual justifications and our research questions:

1. AMD correctly classified with probability close to 1.

2. Normal scan correctly classified with probability close to 1.

3. AMD correctly classified with probability close to 0.5 (mild retinal alteration).

4. Normal scan incorrectly classified as AMD.

5. AMD correctly classified by the multimodal approach, but incorrectly classified by the first the unimodal strategy that considers OCT images only.

6. AMD incorrectly classified as normal by the multimodal approach.

Fig 3 corresponds to a left-eye scan with evident abnormal findings. AMD is correctly classified with probability close to 1.

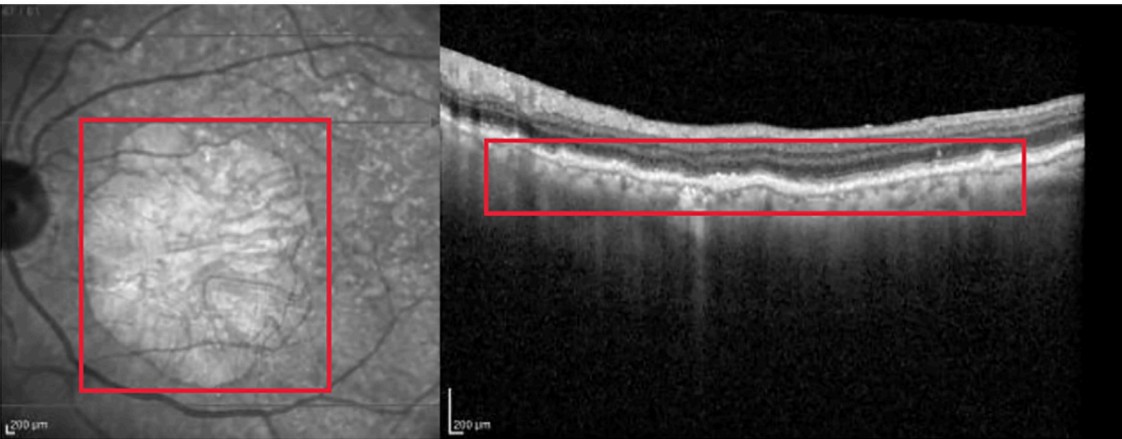

**Fig 3. Evident AMD correctly classified with probability close to 1.** The red rectangle highlights the right area in the OCT image.

Grad-CAM identifies the region of interest correctly for both the IR and OCT images (see Fig 4).

Guided Backpropagation is able to emphasize the damage in the IR image, while delimiting the different profiles in the OCT scan where AMD is located (see Fig 5).

Finally, Fig 6 presents the visualization for the Guided grad-CAM method. Both Guided Backpropagation and Guided grad-CAM provide similar information, but guided grad-CAM is easier to interpret.

Fig 7 presents a scan with no retinal damage.

Grad-CAM highlights the main region of interest correctly for both the IR and OCT images (see Fig 8).

Guided Backpropagation is able to emphasize the layers of the retina in the OCT; however, the IR highlights an artifact (the circle), which is not useful for the analysis (see Fig 9).

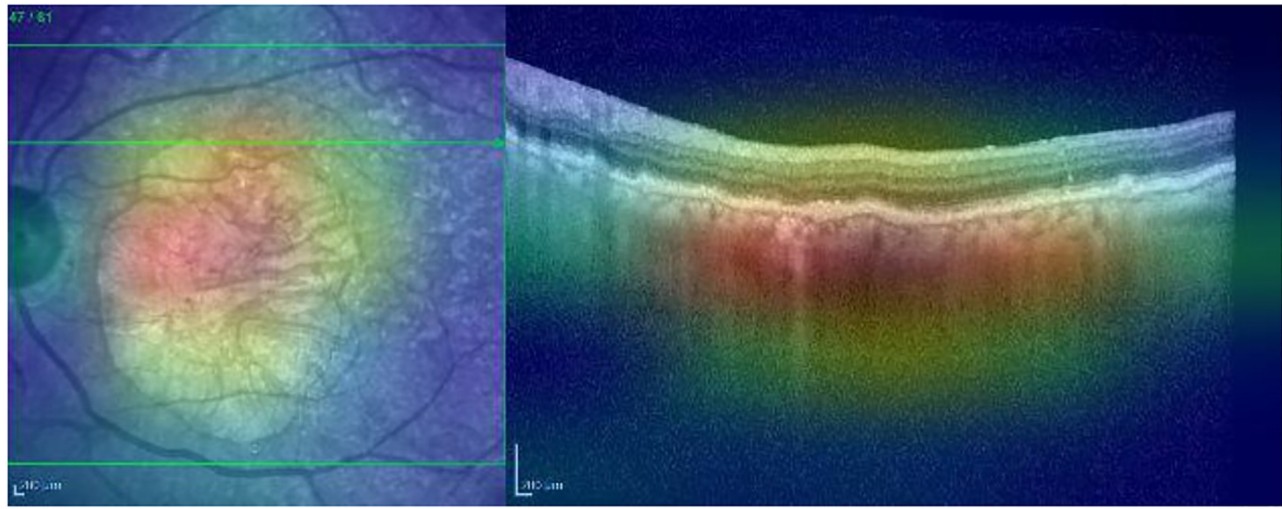

**Fig 4. Grad-CAM heatmap for the AMD case presented in Fig 3.**

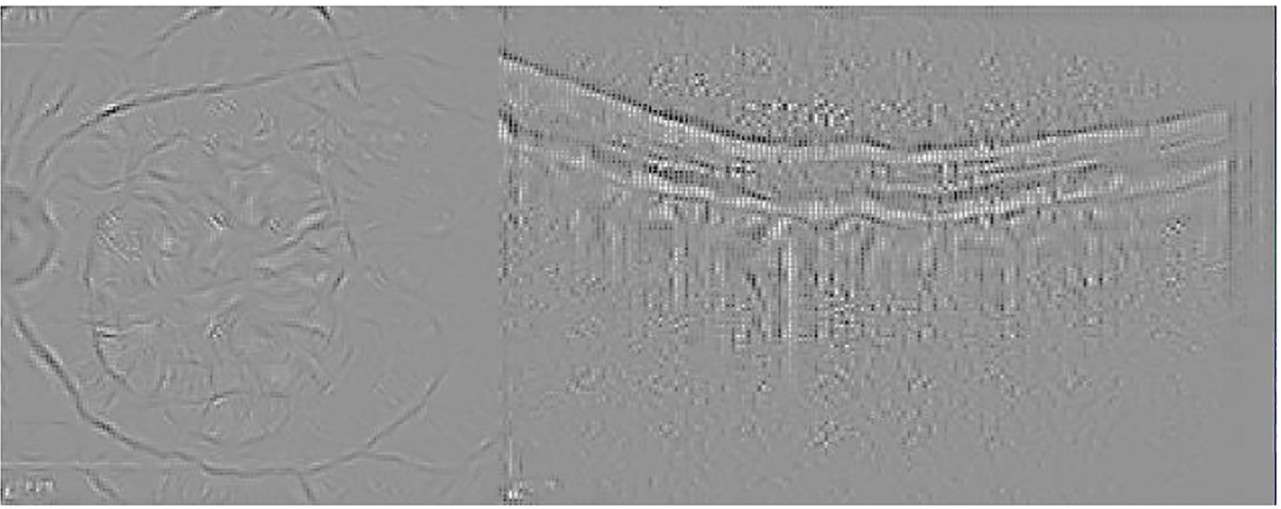

**Fig 5. Guided Backpropagation visualization for the AMD case presented in Fig 3.**

Finally, the Guided grad-CAM visualization in Fig 10 offers an interpretation comparable to that of Guided Backpropagation. As seen in the AMD example in Fig 3, Guided Backpropagation conveys the same information as Guided grad-CAM, but the resulting image is darker and more difficult to interpret.

Fig 11 presents a mild case of retinal damage, which is classified correctly by the model with a probability of 0.698.

Grad-CAM (Fig 12) emphasizes the central area, showing alterations of the external retina. However, the retinal alteration is only partially identified by the model (the drusen -red boundary box in Fig 11 -falls in the orange/red zone).

However, Guided Backpropagation (Fig 13) does not reveal the alteration in the retina.

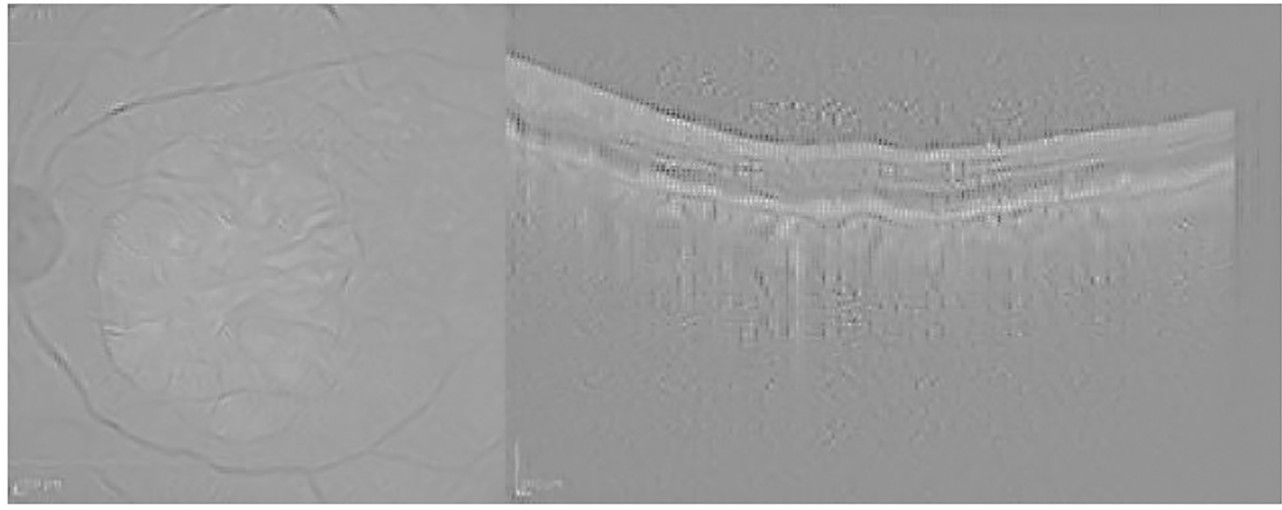

**Fig 6. Guided grad-CAM visualization for the AMD case presented in Fig 3.**

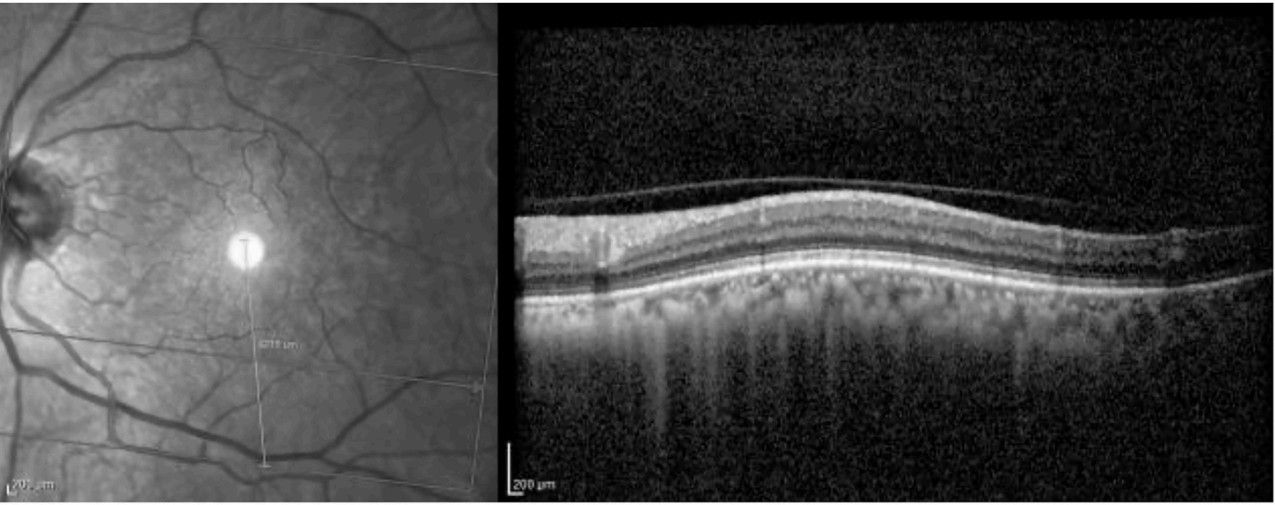

**Fig 7. Normal scan correctly classified with probability close to 1.** The red rectangle highlights the area of interest.

Finally, guided grad-CAM (Fig 14) is able to emphasize the retinal thickness profile, but with difficulty. This is somewhat expected as the model can classify the sample correctly but with a low confidence. In such scenarios, the XAI method merely mirrors the model's inherent uncertainty and does not directly affect its performance.

Fig 15 is a normal scan (no AMD) with non-specific retinal changes that can be observed in the IR and OCT, and the model misclassified this case as AMD.

Grad-CAM (Fig 16) emphasizes the area in the IR and OCT images in which the alteration is present.

Guided Backpropagation (Fig 17) is able to detect this alteration and demarcate the layers of the retina. However, is sensitive to artifacts.

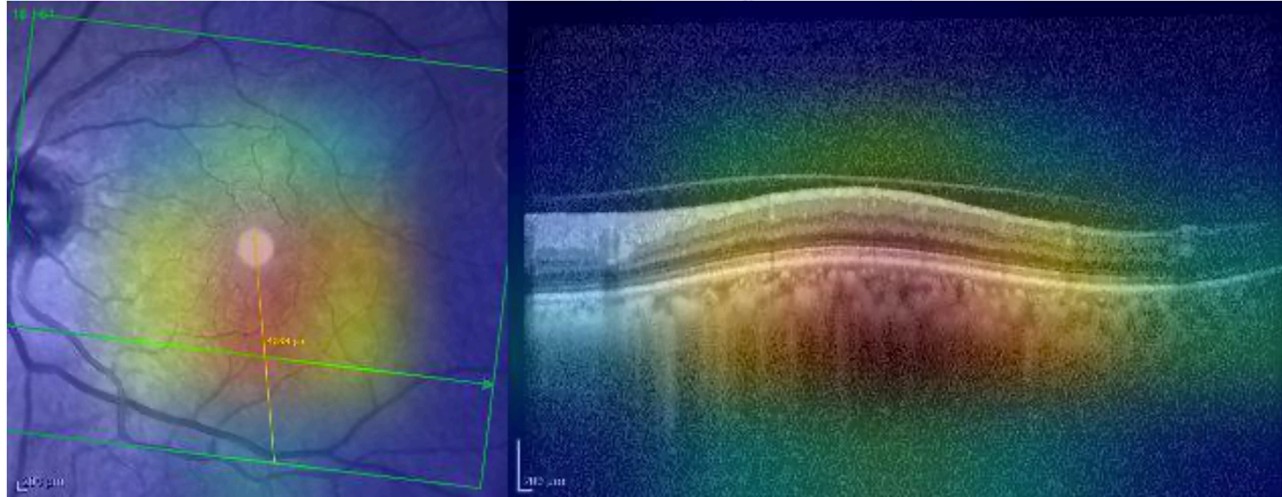

**Fig 8. Grad-CAM heatmap for the normal case presented in Fig 7.**

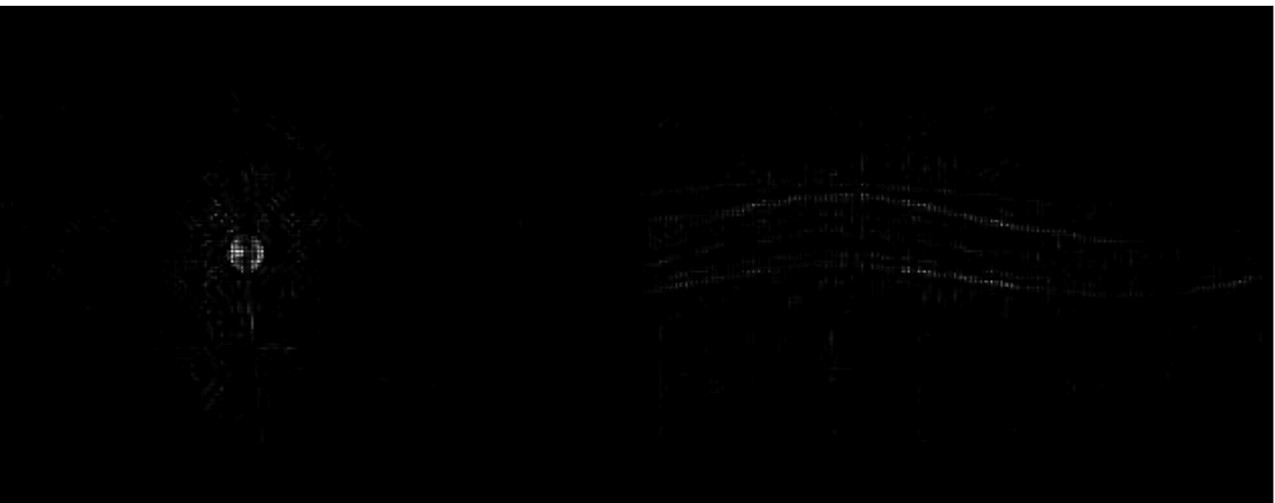

**Fig 9. Guided Backpropagation visualization for the normal case presented in Fig 7.**

Finally, the Guided grad-CAM visualization (Fig 18) has the same interpretation, being slightly less sensitive to artifacts. Grad-CAM and guided grad-CAM identify the area of interest correctly (the red rectangles in Fig 15), even when the image is classified incorrectly.

The next example corresponds to a left-eye AMD scan classified correctly by our multimodal framework, while the unimodal approach based on OCT images classified it as normal. Fig 19 shows the OCT image of the latter approach.

Grad-CAM (Fig 20) highlights a zone in which the retinal damage is not present (the main alteration is located in the yellow area).

Guided Backpropagation (Fig 21) is also not able to detect this alteration.

Similarly, Guided grad-CAM (Fig 22) does not detect the AMD case.

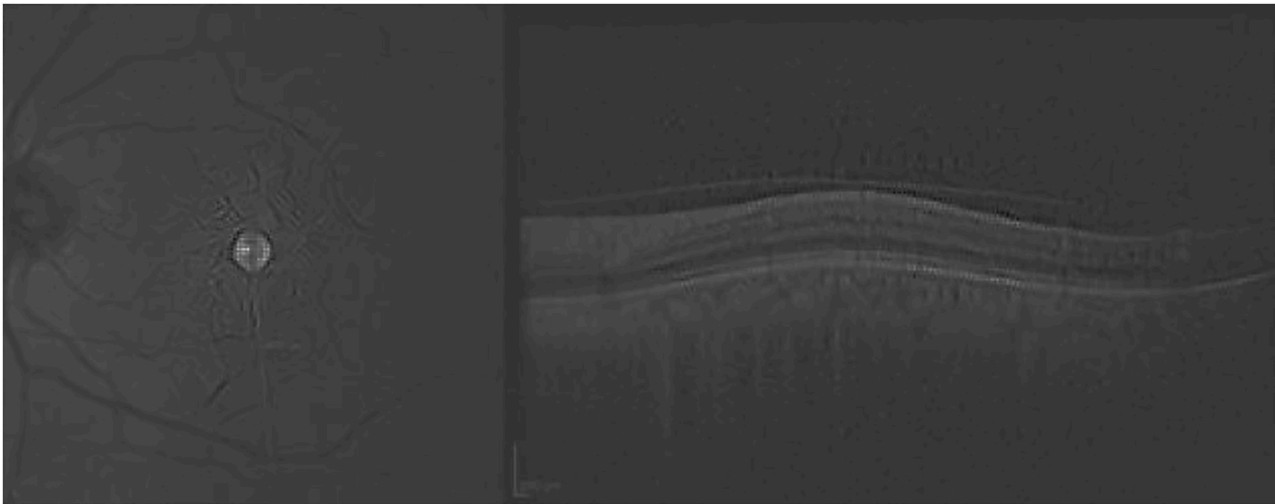

**Fig 10. Guided grad-CAM visualization for the normal case presented in Fig 7.**

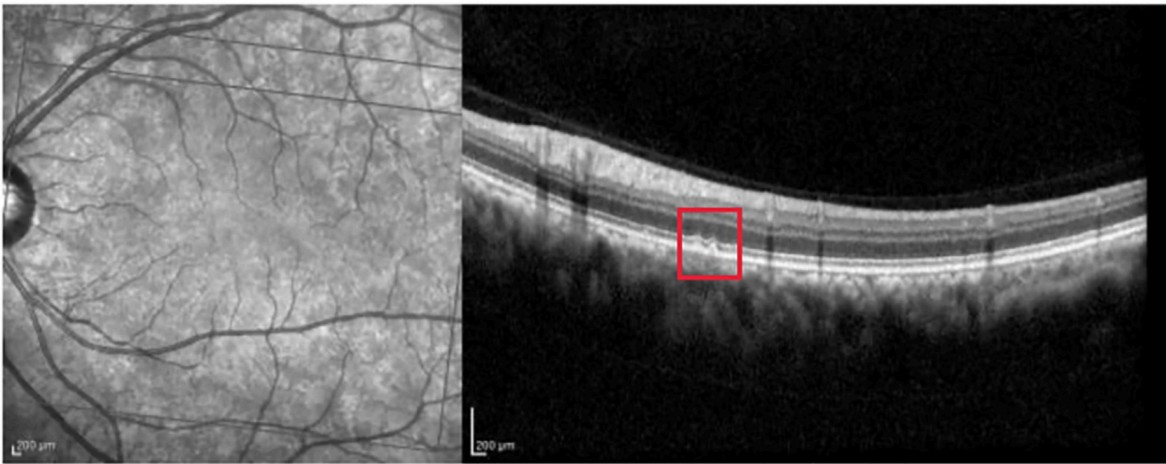

**Fig 11. AMD scan with mild retinal damage.** The red rectangle highlights the drusen in the OCT image.

Fig 23 shows the same left-eye AMD scan presented in the previous figure when analyzed in a multimodal manner. There are hyporeflective spots in the IR (red boundary box at the left-hand side of Fig 23) that justifies the AMD diagnosis. There are very subtle alterations that can be observed in the OCT of the macula (red boundary box at the right-hand side of Fig 23), however, this image is not as informative as the IR.

The punctiform lesions are highlighted adequately by Grad-CAM in the IR (Fig 24). However, this XAI technique is not able to identify the alterations in the OCT.

In contrast to Grad-CAM, Guided Backpropagation (Fig 25) is not able to emphasize the hyporeflective spots in the IR.

Similarly, Guided grad-CAM (Fig 26) is not able to identify the region of interest. Similar to Fig 11, this issue seems to be caused by the inability of the CNN model to identify the retinal alteration in the OCT image rather than an isolated problem with the XAI techniques. This is

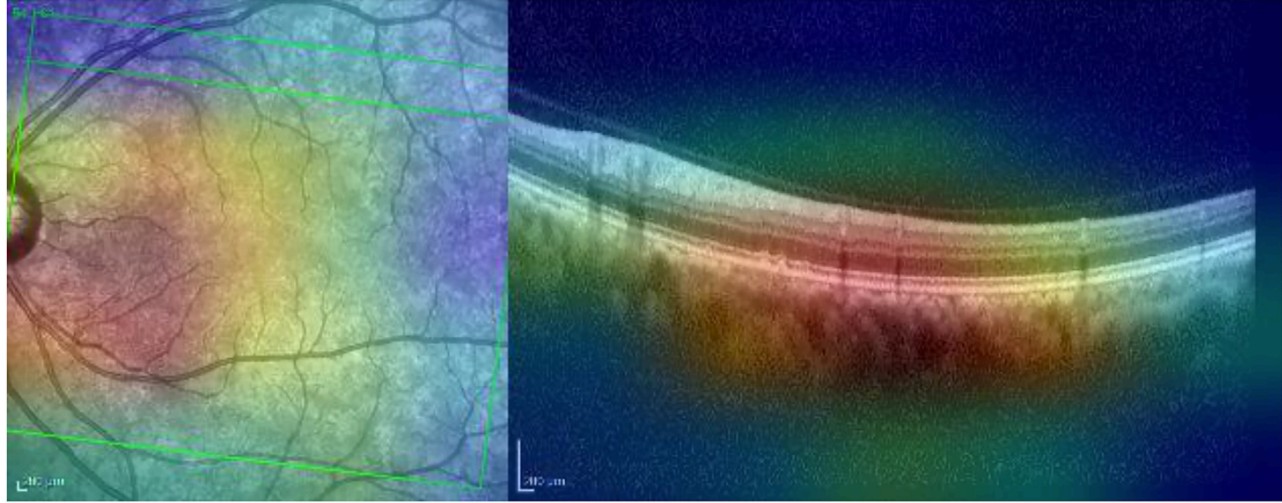

**Fig 12. Grad-CAM heatmap for the mild AMD case presented in Fig 11.**

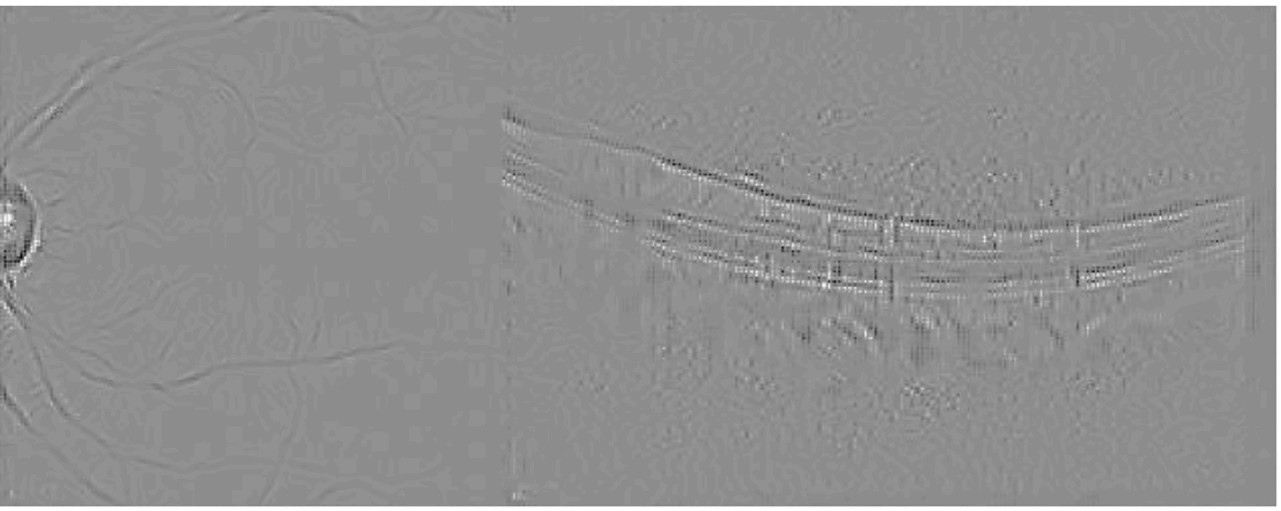

**Fig 13. Guided Backpropagation visualization for the mild AMD case presented in Fig 11.**

confirmed by Fig 19. This is an example of the ability of the multimodal approach to identify the condition (AMD) by adequately weighting information in informative and less informative images. This confirms the good results reported in Table 1.

In summary, the Grad-CAM provides a coarse heatmap in which it is easier to identify retinal damage in relation to the other two methods. Guided Grad-CAM is an interesting second approach, since it provides a fine-grained visual justification that complements very well the Grad-CAM. Finally, Guided backpropagation does not seem to add value in relation to the previous two techniques.

Nevertheless, the explanations presented by the preferred XAI methods (Grad-CAM and Guided Grad-CAM) are not perfect and they should be always used with caution. Several pitfalls with XAI techniques have been acknowledged in the literature (see, e.g., [35]), which are

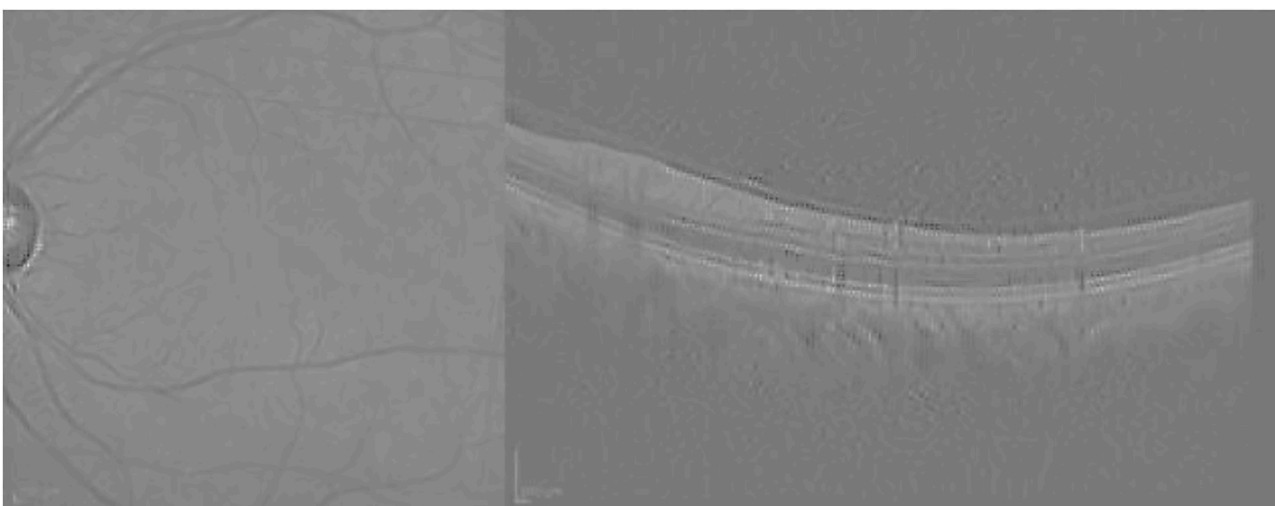

**Fig 14. Guided grad-CAM visualization for the mild AMD case presented in Fig 11.**

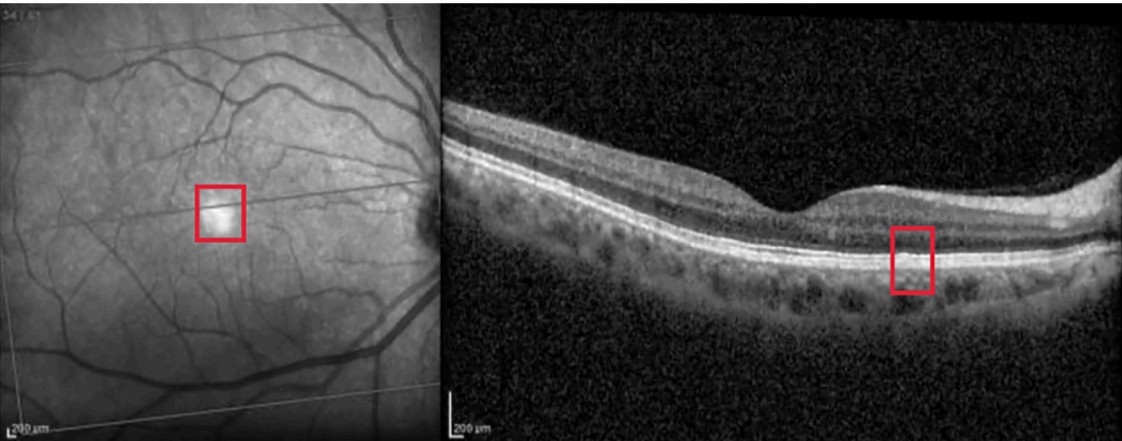

**Fig 15. Normal scan misclassified as AMD (mild alteration but no AMD).**

confirmed in this study. Following [35], inaccurate XAI techniques limit trust in both the explanation provided by them and the black box model by extension. If we cannot know with certainty whether the output of the XAI model is correct, we cannot know whether to trust either the explanation or the machine learning model. XAI techniques can be wrong with patterns correctly identified by the learning machines, but also they can make an accurate visualization of mistakes made by the black-box technique.

As a final example, Fig 27 illustrates an AMD sample that was incorrectly classified by the proposed methodology.

The retinal damage is identified correctly with Grad-CAM in the IR image (Fig 28, left-hand side). However, the B-scan slab in the IR (green arrow in the superior temporal vascular arcade) falls outside the area in which the AMD is observable. Notice that AMD is a pathology that mainly affects the macular area and not extra-arcades as in this case. Therefore, the OCT scan is normal. Although Grad-CAM emphasizes the correct area for a normal diagnosis in

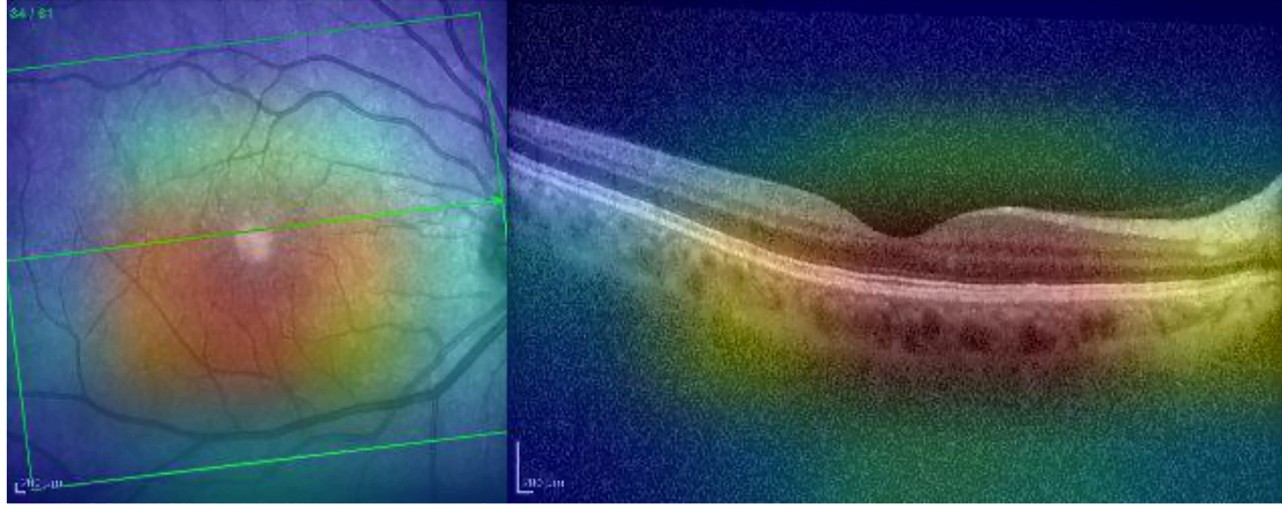

**Fig 16. Grad-CAM heatmap for the normal scan presented in Fig 15.**

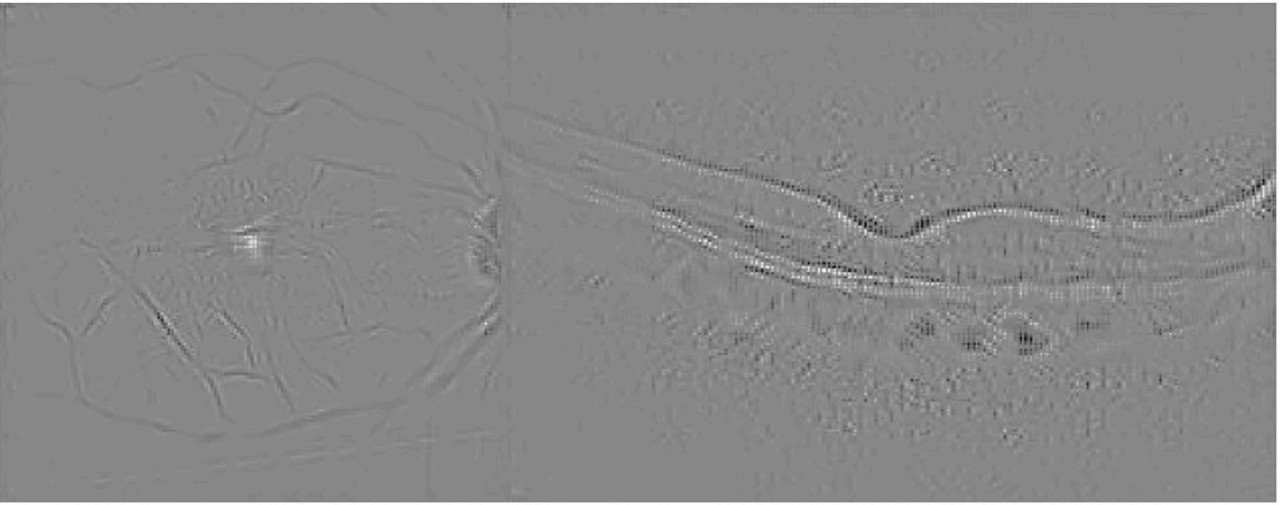

**Fig 17. Guided Backpropagation visualization for the normal scan presented in Fig 15.**

the OCT image (Fig 28, right-hand side), the learning machine fails at balancing these two contradictory outcomes and concludes that the sample is normal.

Guided Backpropagation (Fig 29) is also able to identify the region of interest in the IR and OCT images, although retinal damage is only present in the IR.

Finally, Guided grad-CAM (Fig 30) shows a similar analysis. We conclude that the XAI methods are robust to noisy examples which are not easy to classify.

A second issue indicated by Rudin [35] is that XAI models may not provide enough details to understand how the learning machine classifies. This is particularly true for Guided Grad-CAM and Guided backpropagation, which present fine-grained visualizations. These XAI techniques are harder to interpret as OCT and Fundus imaging are very detailed representations of the eye. As mentioned in Rudin [35], interpretability must be defined in a domain-

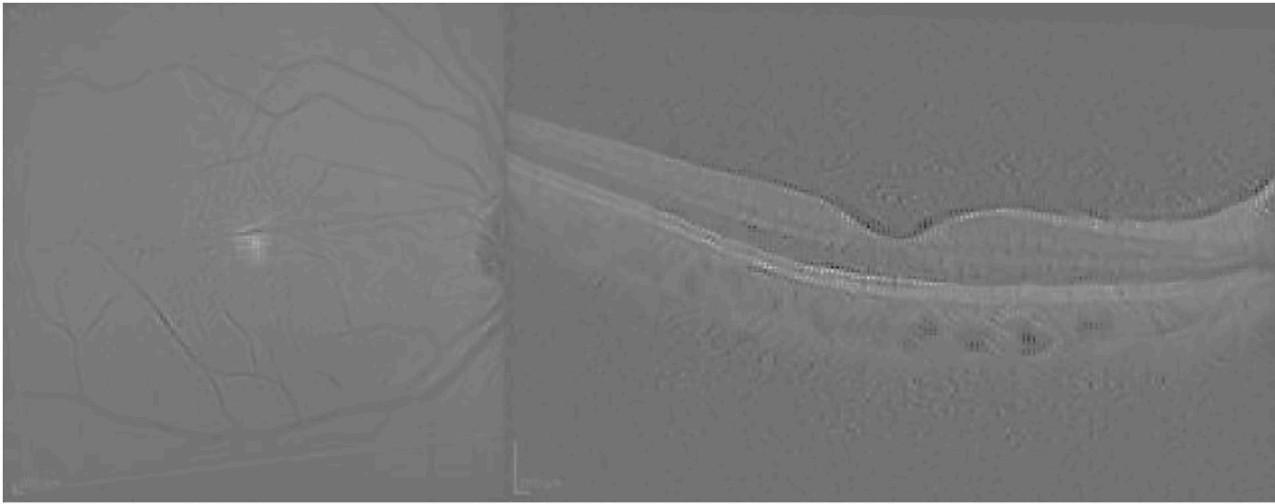

**Fig 18. Guided grad-CAM visualization for the normal scan presented in Fig 15.**

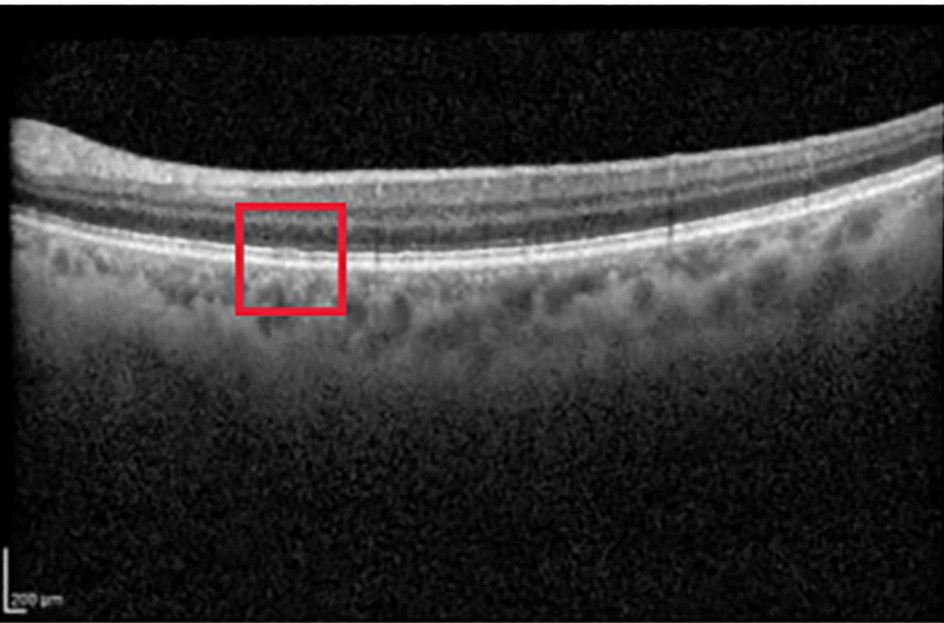

**Fig 19. Scan with AMD misclassified as normal by the unimodal approach based on OCT images only.**

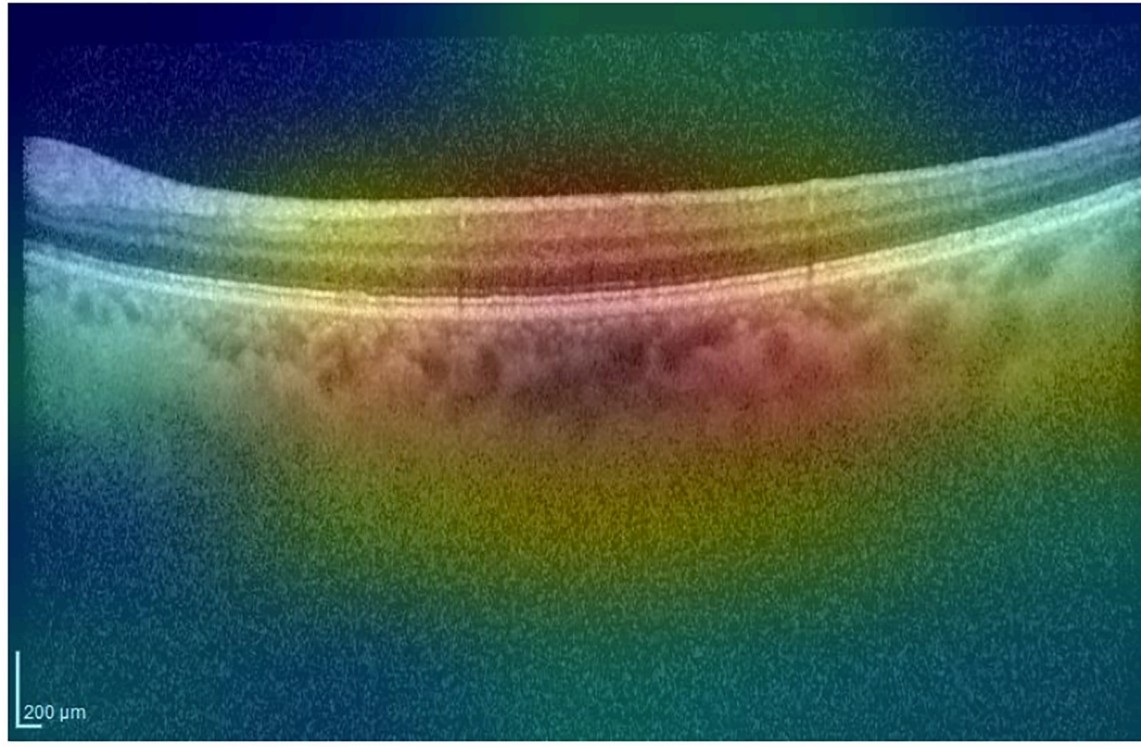

**Fig 20. Grad-CAM heatmap for the AMD scan presented in Fig 19.**

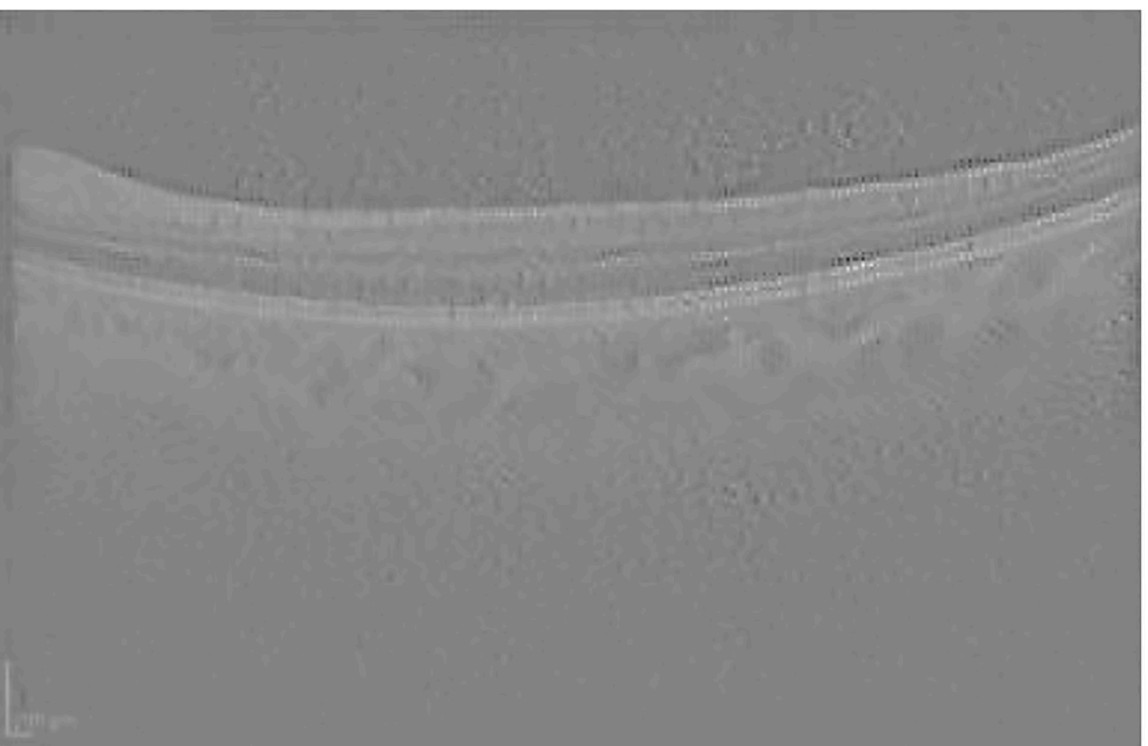

**Fig 21. Guided Backpropagation visualization for the AMD scan presented in Fig 19.**

specific manner, and some methods that work well in some applications may fail in others. The methodological challenges in XAI are related to specific domains, and the methods can be adapted to the nature of the images. Fortunately, the coarse visualization provided by Grad-CAM is clearer and easier to interpret, resulting in a very useful tool for ophthalmologists.

In relation to domain-specific XAI, the goal of this study is to assess the capabilities of XAI methods with experienced ophthalmologists to aid the diagnosis of retinal diseases. The performance of XAI techniques can be assessed from a methodological perspective using different evaluation metrics. Although it would be interesting to see which metrics perform better in terms of robustness in classification, it offers little insights for improving the decision-making process regarding the diagnosis of retinal diseases. For example, guided grad-CAM could perform better than grad-CAM using XAI metrics, however, its fine-grained visualization may not be as useful as grad-CAM for ophthalmologists to reach a better diagnosis.

There are several studies that compare XAI techniques from a methodological perspective (see, e.g., [36]). The explanations can be evaluated by extracting bounding boxes out of the XAI method and comparing them with the real bounding box annotations. We aimed at implementing this strategy in the experimental section, creating red bounding boxes defined by retina specialists. Alternatively, the explanations can be evaluated by using them to perturbate the image, blackening the regions with a low score, and then checking whether the confidence drops or increases when applying the classifier in relation to the original image [36].

## Conclusions

In this work, we propose a novel framework for AMD diagnosis using multimodal deep learning and XAI techniques. We employ a standard XAI strategy, training two CNNs with late

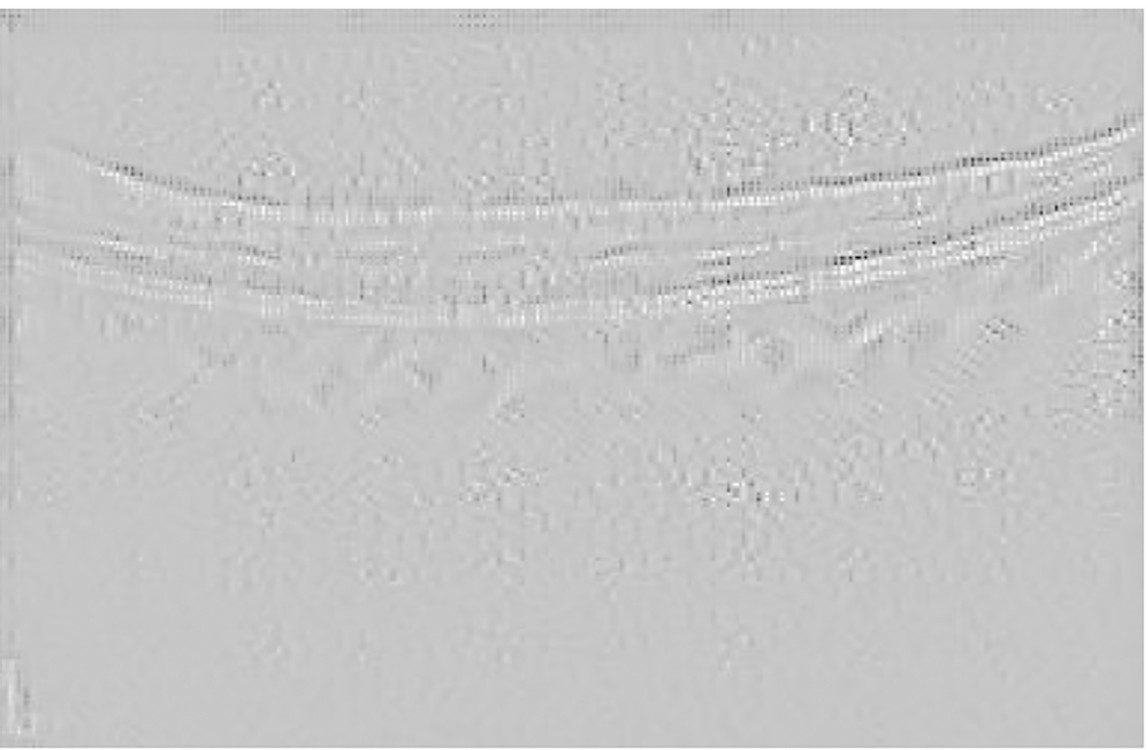

**Fig 22. Guided grad-CAM visualization for the normal scan presented in Fig 19.**

fusion, and implementing a system that provides visual justifications for outcomes (normal or retinal damage) using various visualization approaches. We present seven XAI examples that highlight the most relevant instances in the context of visual justifications.

In terms of predictive performance, our best model achieves a 0.98 AUC and a 94% accuracy. These results are excellent and comparable with the existing literature. For example,

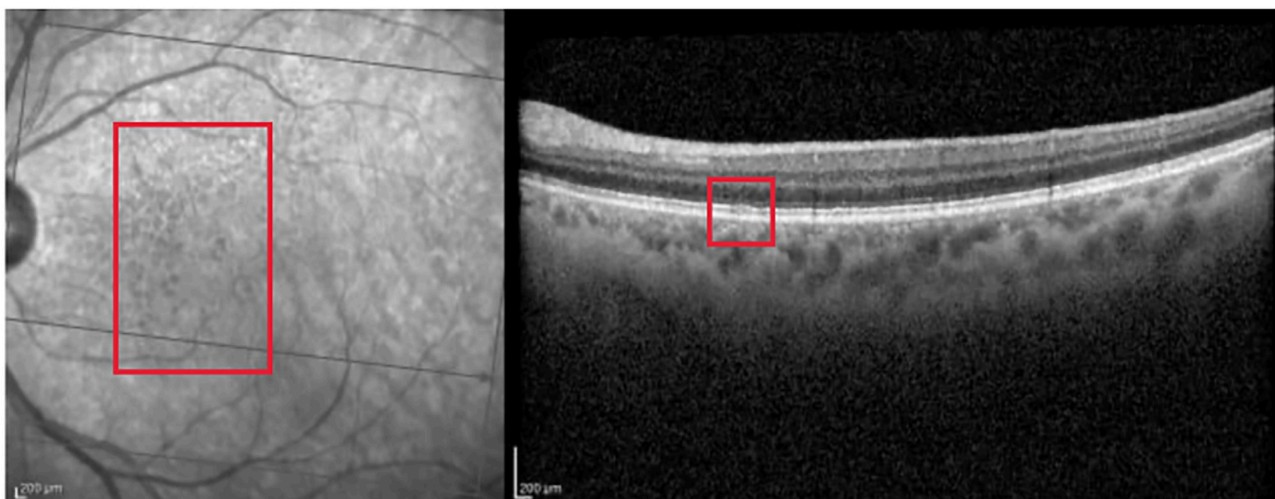

**Fig 23. AMD scan classified correctly by the multimodal approach, but misclassified as normal by the unimodal methods.**

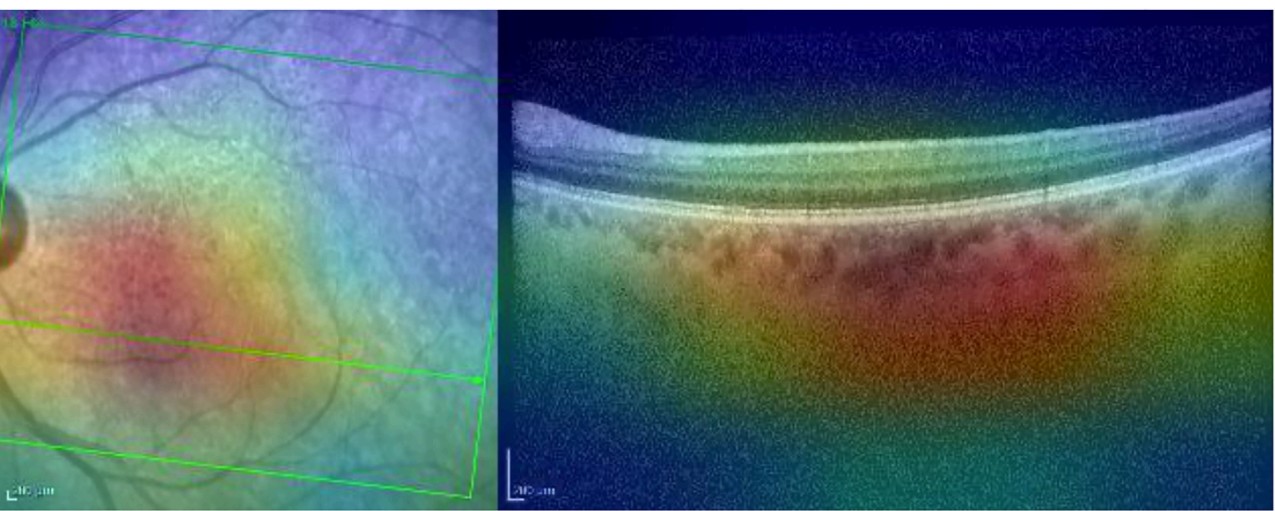

**Fig 24. Grad-CAM heatmap for the AMD scan presented in Fig 23.**

previous studies report relatively similar results; AUC = 0.94 [3], AUC = 0.98 [15], AUC = 0.99 [13], F1 = 0.99 [11, 12, 14].

The two main XAI strategies used in this paper are Grad-CAM and Guided Grad-CAM. Grad-CAM offers a coarse visual justification, highlighting regions where retinal damage may be located, such as the central zone of the macula or fovea centralis. In contrast, Guided Grad-CAM provides a fine-grained visualization of the different retinal layers. This two-step analysis mirrors the diagnostic process of an ophthalmologist: first examining the area of interest, then inspecting the retinal layers. Thus, the proposed visual tool, based on these XAI techniques, is highly useful as it aligns with the typical AMD diagnostic process.

Based on our analysis, we can provide an answer to the research questions:

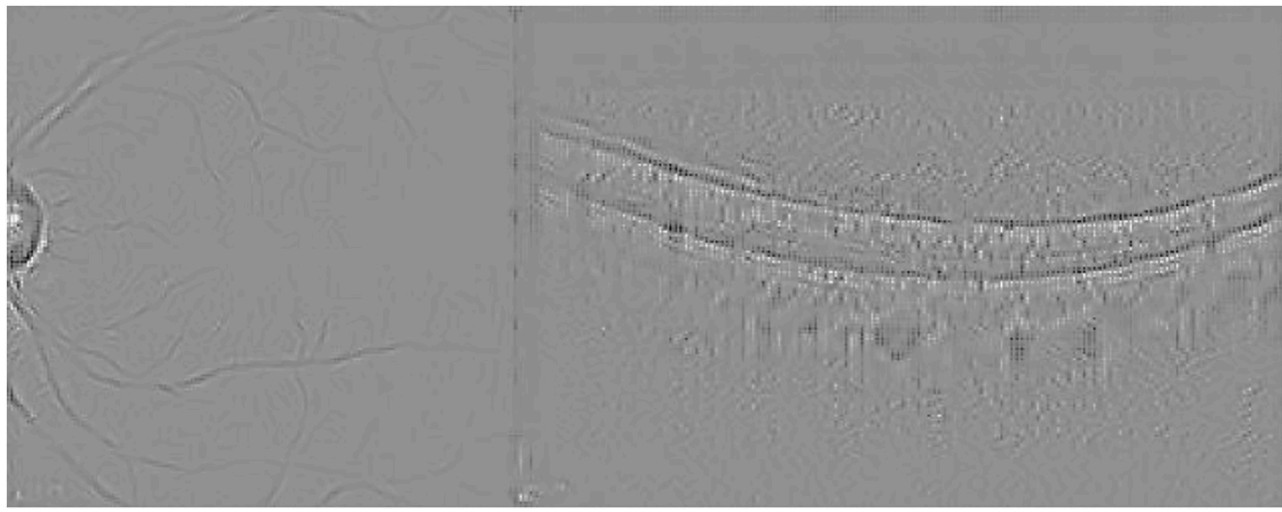

**Fig 25. Guided Backpropagation visualization for the AMD scan presented in Fig 23.**

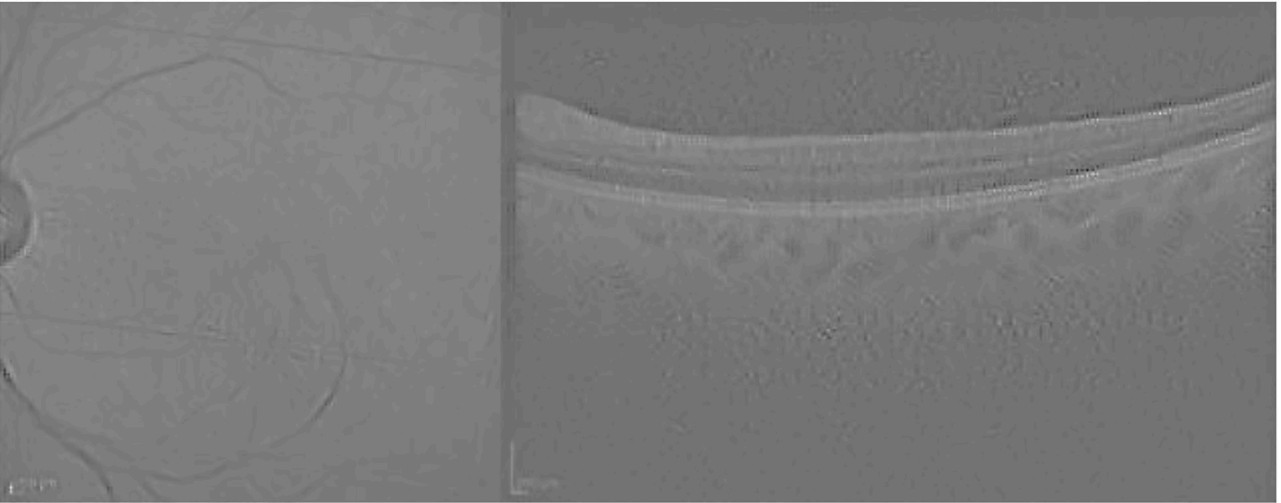

**Fig 26. Guided grad-CAM visualization for the normal scan presented in Fig 23.**

**RQ1:** Is IR a valuable data source for AMD prediction? Yes, the use of IR leads to best predictive results when a suitable multimodal architecture is considered.

**RQ2:** Does the generation of visual justifications for both IR and OCT imaging leads to a better interpretation for AMD diagnosis? Yes, the XAI methods grad-CAM and guided grad-CAM were successful at identifying the regions of interest in both images, even when the scans were misclassified. The proposed framework was validated by ophthalmologists, and considered as a valuable tool for AMD diagnosis.

**RQ3:** Does the combination of two or more XAI methods provides a better interpretation for AMD diagnosis in relation to a single strategy? Yes, grad-CAM provides a coarse location of the zone of interest where AMD is supposed to be located, while guided grad-CAM

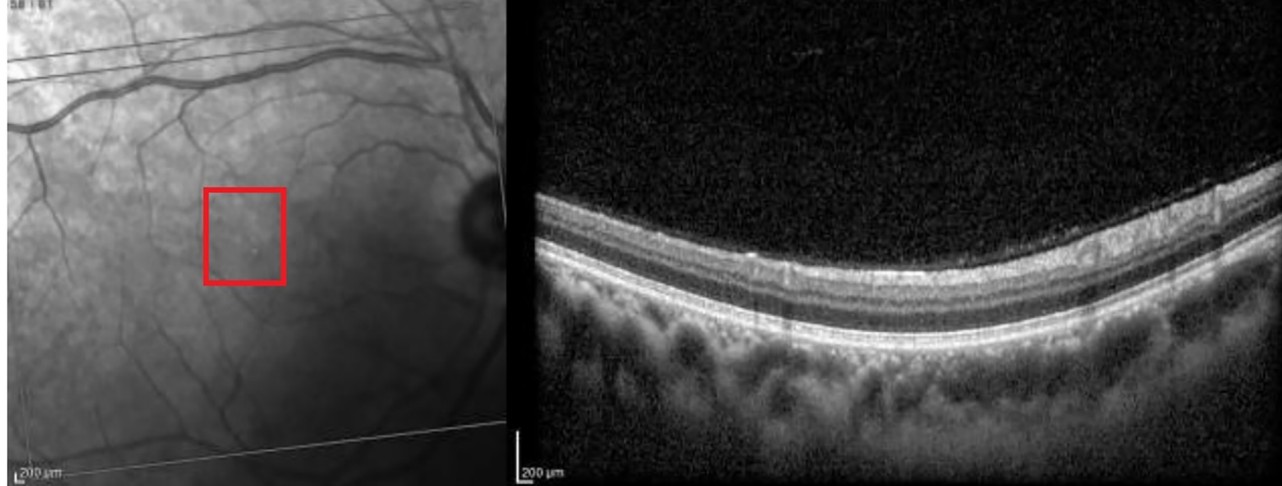

**Fig 27. AMD scan classified incorrectly by the multimodal approach.**

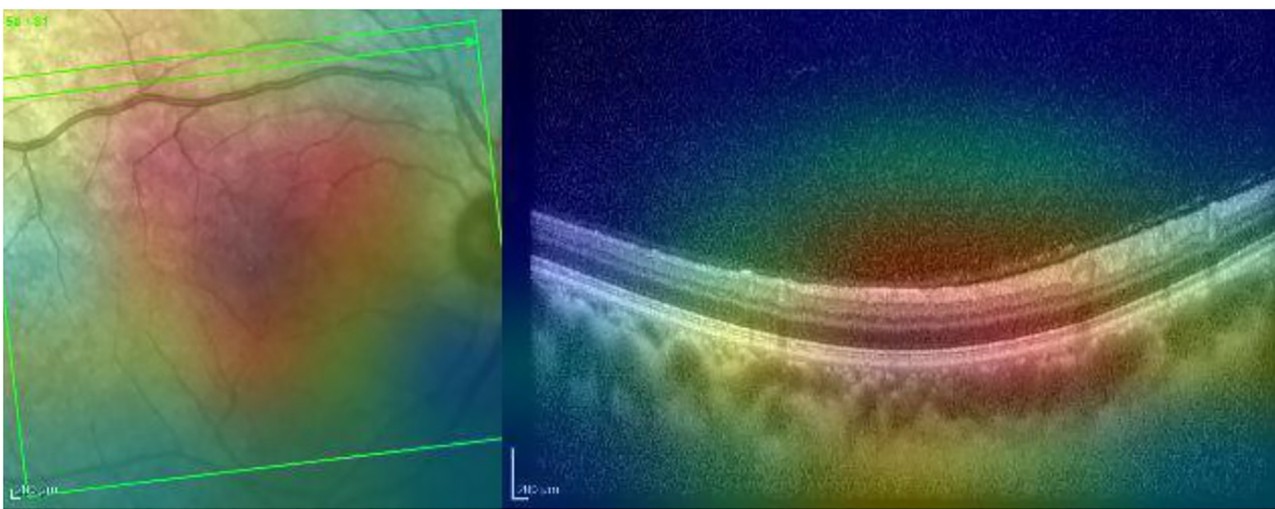

**Fig 28. Grad-CAM heatmap for the AMD scan presented in Fig 27.**

generates a fine grained visualization, which is useful for analyzing the retinal layers. The two XAI methods can be applied to both OCT and IR imaging. The four visual justifications provide a diagnosis tool that is in line with the procedure followed by ophthalmologists.

Our contribution is mainly applied as this is the first XAI study that combines OCT and IR imaging in a multimodal fashion. We show empirically the importance of IR imaging since it provides a general overview of the macula, complementing the information of the OCT image. We can observe in the IR image subretinal alterations in the retinal photoreceptor layer, the choroid, and the retinal pigmented epithelium (RPE) [37].

In terms of practical implications, the proposed system can be integrated into clinical practice to perform automatic analysis of OCT/IR imaging. The system can provide insights that may be difficult for ophthalmologists to identify on their own. To ensure successful

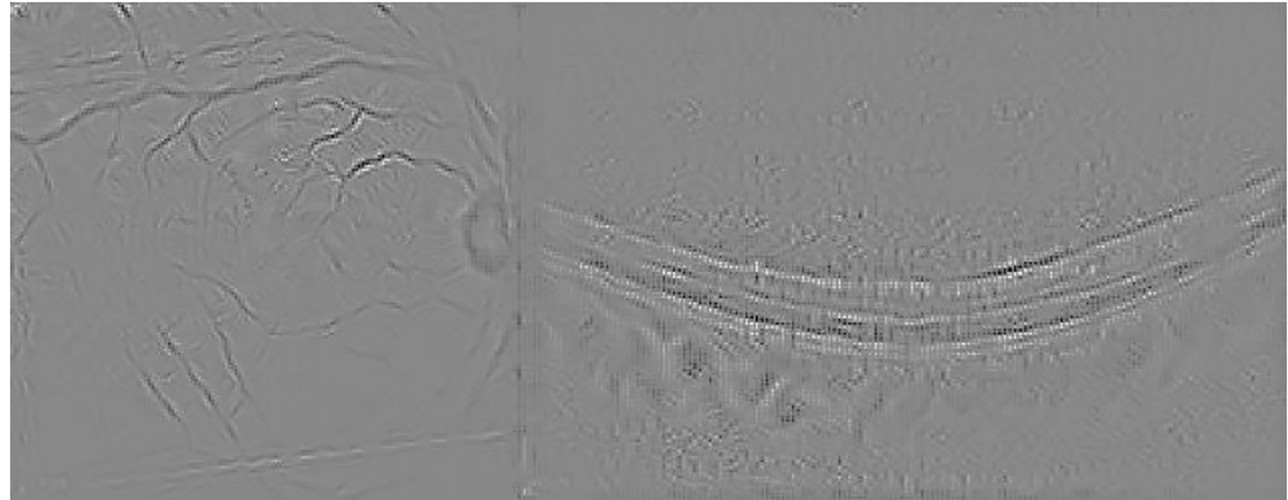

**Fig 29. Guided Backpropagation visualization for the AMD scan presented in Fig 27.**

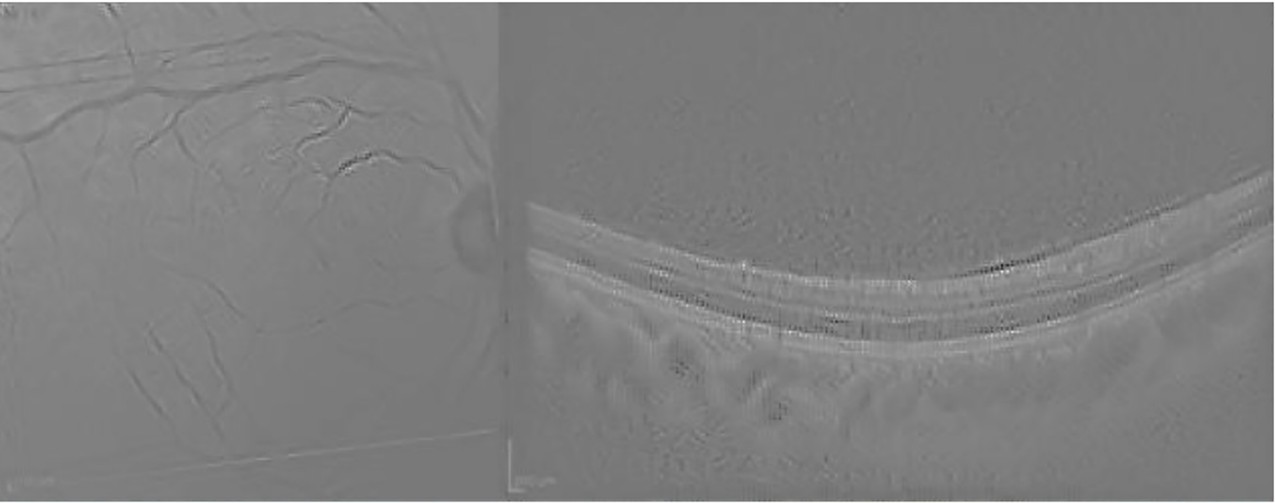

**Fig 30. Guided grad-CAM visualization for the normal scan presented in Fig 27.**

integration, it is essential to adopt a collaborative approach where AI tools augment, rather than replace, the expertise of healthcare professionals. This requires proper training, ongoing evaluation of AI performance, and a robust infrastructure to support the seamless incorporation of AI into existing clinical workflows.

The main limitation of the framework regarding its integration into clinical practice is that it generates four new images based on XAI methods: the OCT and IR images along with the outcomes of the Grad-CAM and Guided Grad-CAM methods. This may result in an overwhelming amount of information for the ophthalmologist, even with the removal of Guided Backpropagation. As future work, we intend to enhance this framework by ranking the XAI images based on their ability to aid in diagnosis through an automated process. For a given patient, the optimal combination of visual justifications might be the Grad-CAM output for the IR image and the Guided Grad-CAM output for the OCT image, for example, while different combinations of images could be more suitable in other cases.

Another limitation of this study is the use of default configurations of well-known CNN classifiers without modifying their structure. Alternatively, ad-hoc bimodal architectures can be defined to improve the accuracy of the model. We consider this option as an interesting future research avenue.

## Author Contributions

**Conceptualization:** Sebastián Maldonado, Cristhian A. Urzua.

**Data curation:** Loreto Cuitino, Cristhian A. Urzua.

**Formal analysis:** Loreto Cuitino.

**Investigation:** Loreto Cuitino.

**Methodology:** Carla Vairetti.

**Project administration:** Cristhian A. Urzua.

**Software:** Carla Vairetti, Sebastián Maldonado.

**Writing – original draft:** Carla Vairetti, Sebastián Maldonado, Loreto Cuitino, Cristhian A. Urzua.

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
