## [Decision Letter · Decision Letter 0]

26 Jul 2024

PONE-D-24-15878Interpretable Multimodal Classification for Age-related Macular Degeneration DiagnosisPLOS ONE

Dear Dr. Vairetti,

Thank you for submitting your manuscript to PLOS ONE. After careful consideration, we feel that it has merit but does not fully meet PLOS ONE’s publication criteria as it currently stands. Therefore, we invite you to submit a revised version of the manuscript that addresses the points raised during the review process. 

We look forward to receiving your revised manuscript.

Kind regards,

Muhammad Mateen

Academic Editor

PLOS ONE

 [The authors gratefully acknowledge financial support from ANID PIA/PUENTE AFB230002 and FONDECYT-Chile, grants 12200007, 1200221, 11191215, and 1212038.].  

4. In the online submission form, you indicated that [Data cannot be shared publicly because it belongs to the Hospital Clínico Universidad de Chile, Santiago, Chile. The data can be shared privetely upon reasonable request.]. 

5. We note that Figure 3,4,5,6,7,8, and 9 in your submission contain [map/satellite] images which may be copyrighted. All PLOS content is published under the Creative Commons Attribution License (CC BY 4.0), which means that the manuscript, images, and Supporting Information files will be freely available online, and any third party is permitted to access, download, copy, distribute, and use these materials in any way, even commercially, with proper attribution. For these reasons, we cannot publish previously copyrighted maps or satellite images created using proprietary data, such as Google software (Google Maps, Street View, and Earth). For more information, see our copyright guidelines: http://journals.plos.org/plosone/s/licenses-and-copyright.

1. You may seek permission from the original copyright holder of Figure 3,4,5,6,7,8, and 9   to publish the content specifically under the CC BY 4.0 license.  

Additional Editor Comments:

The authors need to address the reviewer's comments carefully. It is also recommended to cite the few relevant recent studies of current year 2024.

Reviewers' comments:

Reviewer's Responses to Questions

**Comments to the Author**

1. Is the manuscript technically sound, and do the data support the conclusions?

Reviewer #1: Partly

Reviewer #2: Yes

2. Has the statistical analysis been performed appropriately and rigorously? 

Reviewer #1: No

Reviewer #2: Yes

3. Have the authors made all data underlying the findings in their manuscript fully available?

Reviewer #1: Yes

Reviewer #2: No

4. Is the manuscript presented in an intelligible fashion and written in standard English?

Reviewer #1: Yes

Reviewer #2: Yes

5. Review Comments to the Author

Reviewer #1: Comment 1

• Novelty of combining optical coherence tomography (OCT) and infrared reflectance (IR) imaging is mentioned, but the paper does not sufficiently compare this with existing methods or justify the advancements over prior approaches.

• Provide a comparative analysis of your method with recent studies in multimodal imaging for AMD and explain specifically how your approach advances the field. Include relevant recent literature to strengthen the argument.

Comment 2

• The introduction is verbose and could be more concise.

• Streamline the introduction to focus on the key points: the clinical importance of AMD, challenges in diagnosis, and how your approach addresses these challenges. Avoid redundant explanations and ensure a clear, direct narrative.

Comment 3

• The datasets used are not described in enough detail.

• Provide comprehensive details about the datasets, including the number of images, the source, inclusion/exclusion criteria, and any preprocessing steps. Discuss the representativeness of the datasets and potential biases.

Comment 4

• The training and validation process of the models lacks detail.

• Elaborate on the model training process, including data partitioning (e.g., train/test split), any cross-validation techniques used, hyperparameter tuning, and criteria for model selection. Include a flowchart or pseudocode if possible.

Comment 5

• The use of Grad-CAM, guided backpropagation, and guided Grad-CAM is described but could be clearer.

• Provide a more detailed explanation of how these techniques work and their implementation in your model. Include diagrams or visual examples to illustrate how these techniques contribute to model interpretability.

Comment 6

• Results are primarily reported in terms of accuracy.

• Include additional performance metrics such as precision, recall, F1-score, and AUC-ROC to provide a more comprehensive evaluation of your model's performance. Present these in a table for clarity and comparison.

Comment 7

• The results lack detailed statistical analysis.

• Conduct statistical tests to determine the significance of your findings. Report confidence intervals and p-values where appropriate. Discuss the variability and robustness of your results.

Comment 8

• The interpretability results are not validated with real-world feedback.

• Consider conducting user studies or obtaining feedback from ophthalmologists to validate the interpretability of the explanations provided by your model. Include their perspectives in the discussion.

Comment 9

• The discussion lacks depth regarding practical implications.

• Discuss how your system could be integrated into clinical practice, potential challenges in deployment, and how it might impact clinical workflows. Address any limitations and suggest ways to overcome them.

Comment 11

• Figures and tables could be improved to better illustrate key points.

• Ensure all figures and tables are well-labeled and add explanatory captions. Use visual examples to show how the interpretability techniques work on specific cases of AMD. Consider adding a figure that summarizes the workflow of your method.

Comment 12

• The conclusion does not clearly summarize the specific contributions and implications.

• Provide a more definitive conclusion that summarizes the key contributions of your work, its impact on the field, and specific advancements over prior methods. Outline clear directions for future research and how your findings can be further validated and applied.

Comment 13

• There are instances of verbose language and potential typos.

• Conduct a thorough proofreading to correct any typos and improve language clarity. Consider using more concise phrasing where possible to enhance readability.

Comment 14

• The references section may not include the latest relevant research.

• Update the references to include recent literature relevant to your study, particularly in the fields of XAI, multimodal imaging, and AMD diagnosis. Ensure all citations are current and properly formatted.

Reviewer #2: I believe this is a very good study with the potential to help improve the use of AI in diagnosis of retinal diseases. The authors have done a good job with a fairly large sample size, they have also highlighted the strengths and weaknesses of the study and also areas for further research.

6. PLOS authors have the option to publish the peer review history of their article (what does this mean?). If published, this will include your full peer review and any attached files.

Reviewer #1: No

Reviewer #2: No

---

## [Author Response · Author response to Decision Letter 0]

23 Aug 2024

[See attached files for formatted responses]

General Comment Journal requirements: 

 [The authors gratefully acknowledge financial support from ANID PIA/PUENTE AFB230002 and FONDECYT-Chile, grants 12200007, 1200221, 11191215, and 1212038.]. 

4. In the online submission form, you indicated that [Data cannot be shared publicly because it belongs to the Hospital Clínico Universidad de Chile, Santiago, Chile. The data can be shared privetely upon reasonable request.]. 

5. We note that Figure 3,4,5,6,7,8, and 9 in your submission contain [map/satellite] images which may be copyrighted. All PLOS content is published under the Creative Commons Attribution License (CC BY 4.0), which means that the manuscript, images, and Supporting Information files will be freely available online, and any third party is permitted to access, download, copy, distribute, and use these materials in any way, even commercially, with proper attribution. For these reasons, we cannot publish previously copyrighted maps or satellite images created using proprietary data, such as Google software (Google Maps, Street View, and Earth). For more information, see our copyright guidelines: http://journals.plos.org/plosone/s/licenses-and-copyright.

- You may seek permission from the original copyright holder of Figure 3,4,5,6,7,8, and 9 to publish the content specifically under the CC BY 4.0 license. 

- If you are unable to obtain permission from the original copyright holder to publish these figures under the CC BY 4.0 license or if the copyright holder’s requirements are incompatible with the CC BY 4.0 license, please either i) remove the figure or ii) supply a replacement figure that complies with the CC BY 4.0 license. Please check copyright information on all replacement figures and update the figure caption with source information. If applicable, please specify in the figure caption text when a figure is similar but not identical to the original image and is therefore for illustrative purposes only.

Additional Editor Comments:

The authors need to address the reviewer's comments carefully. It is also recommended to cite the few relevant recent studies of current year 2024.

Author Answer Thank you very much for your valuable feedback. Below, we respond to each of your questions:

1. We have adjusted the format of the figures, numbering them from Fig1 to Fig30. We have also provided the .tiff files for each of the figures. Please note that the entire manuscript is formatted in LaTeX using the PLOS One template.

2. We agree with the statement. The code will be uploaded to a public repository following the journal’s guidelines in case of publication.

3. The funders mentioned in the original submission had no role in study design, data collection and analysis, decision to publish, or preparation of the manuscript. We have adjusted the acknowledgments accordingly.

4. Our data cannot be made publicly available for ethical and legal reasons. Public availability would compromise patient privacy, and public deposition would breach compliance with the protocol approved by our research ethics board (the original files and their translations were attached in the previous submission). Therefore, we request an exemption for this study in this cover letter.

5. This submission does not include map/satellite images or any copyrighted material. All the images are either diagrams designed for this study or OCT images collected by our research team in accordance with the agreements with the Ethics Board.

Regarding the final comment:

1. We have carefully addressed all the reviewers' comments, which are highly appreciated.

2. Recent references have been included in the revised version of the manuscript.

Reviewer #1

Comment 1.1 

• Novelty of combining optical coherence tomography (OCT) and infrared reflectance (IR) imaging is mentioned, but the paper does not sufficiently compare this with existing methods or justify the advancements over prior approaches.

• Provide a comparative analysis of your method with recent studies in multimodal imaging for AMD and explain specifically how your approach advances the field. Include relevant recent literature to strengthen the argument.

Author Answer Thank you very much for your valuable feedback. Next, we respond each one of your questions.

Author Action We have extended the discussion on multimodal imaging for AMD diagnosis, highlighting the novelty and contribution of our study in relation to recent studies.

The following paragraphs were adjusted/included in Section “Theoretical background on multimodal OCT and XAI techniques”:

Several papers have proposed a multimodal approach to AMD detection. Among the seminal studies, Vaghefi et al. [14] combined OCT and CFP imaging using Inception-ResNet-v2. Alternatively, Yoo et al. [15] also combined OCT and fundus imaging using restricted Boltzmann machine (RBM) and deep belief network (DBN). Although both studies conclude that the combination of different sources is better than a single image, their main issue is the sample size: n=75 and n=83 for Vaghefi et al. [14] and Yoo et al. [15], respectively. A larger study (n=2450) was designed by Keenan et al. [3], which combined FAF and CFP to detect reticular pseudodrusen (RPD). This approach, however, combines images of a similar nature (fundus imaging), excluding OCT samples from the analysis.

In recent years, research on multimodal AMD diagnosis has focused on the analysis of different imaging modalities and AMD variants, such as dry AMD [8]. Wang et al. [16] presented a multimodal approach to AMD detection using OCT and CFP imaging. Anegondi et al. [6] utilized multimodal deep learning to predict the annualized growth rate of geographic atrophy (GA) using OCT and FAF imaging. GA was also analyzed by Winkler et al. [2] using FAF and fundus photos (FP). Oncel et al. [5] examined the relationship between intraretinal hyperreflective foci (IHRF) identified on optical OCT B-scans and hyperpigmentation on CFP. Finally, Goh et al. [1] showed the benefits of multimodal imaging over CFP alone by combining multiple sources such as OCT, FAF, NIR, and CFP.”

We have also improved the discussion on multimodal AMD diagnosis in the introductory section.

Comment 1.2 

• The introduction is verbose and could be more concise.

• Streamline the introduction to focus on the key points: the clinical importance of AMD, challenges in diagnosis, and how your approach addresses these challenges. Avoid redundant explanations and ensure a clear, direct narrative.

Author Answer We agree. Thank you for this suggestion.

Author Action We have adjusted the introduction to present a clear and concise motivation for the manuscript, highlighting the main challenges in AMD diagnosis and providing a clear novelty and contribution statement. 

We have also updated the references on multimodal AMD diagnosis in both the introduction and the prior work section, including four new studies from 2024.

Comment 1.3 

• The datasets used are not described in enough detail.

• Provide comprehensive details about the datasets, including the number of images, the source, inclusion/exclusion criteria, and any preprocessing steps. Discuss the representativeness of the datasets and potential biases.

Author Answer Thank you for this remark. We have provided all the information requested by you. 

Author Action We have included additional information on the dataset, discussing the number of images, the source, inclusion/exclusion criteria, and any preprocessing steps. Discuss the representativeness of the datasets and potential biases.

The following paragraphs were adjusted/included in Section “The dataset and experimental setting”: 

We collected 4563 images of patients seen at the Hospital Clínico Universidad de Chile in Santiago, Chile. This dataset was obtained by scholars and practitioners of the Department of Ophthalmology of the University of Chile. [number of images and source].

Each record consists in the combination of an IR and an OCT image extracted from across the macula cube area. The device used to acquire the images corresponds to a Spectralis OCT 2 imaging platform (Heidelberg Engineering, Germany). [source]

Each scan was manually labeled as `AMD' (dry AMD or wet AMD), `NORMAL', `OTHER', `UNDEFINED' by two experienced retinal specialists with proper training and clinical experience. If the annotations from the two specialists did not coincide, the images were categorized as `UNDEFINED' and subsequently excluded from the analysis. Additionally, images that presented alterations that may prevent the correct diagnostic interpretation by a trained specialist were also labeled as `UNDEFINED'. These artifacts may be due to errors in the software (misidentification of retinal layers, mirror artifact, cut edge artifact), by the operator (degraded image scan, out of register artifact, off center artifact) or by the patient (motion artifact, off center artifact, degraded image scan, mirror artifact) [30, 31]. On the other hand, OCT images with a signal strength index (SSI) lower than 7 were excluded from analysis. [pre-processing and exclusion criteria]

The category `OTHER' represents findings that may be found in diseases other than AMD [32], such as: alterations related to vitreo-retinal interface (vitreo-macular traction, lamellar and full-thickness macular hole, epiretinal membrane), suggestive signs of macular edema (intraretinal hyporeflective spaces) or vascular exudation (intraretinal hipereflective dots), retinal/RPE detachments and alterations in normal macular structure (pathologic myopia). [pre-processing and exclusion criteria]

Records labeled as `OTHER’ or `UNDEFINED’ were discarded, totaling 237, resulting in a final dataset comprised of 4326 images. These exclusions enable the machine to learn from a more refined dataset, thereby reducing model noise. It is important to note that OCT/Fundus images were selected from patients with a potential AMD diagnosis and do not represent all patients with retinal pathologies treated at the Hospital. Images from healthy controls were individually selected to match the age and sex of each of the patients with potential AMD. However, this process was carried out before applying the exclusion criteria, leading to a greater number of normal cases than AMD samples in the final dataset. [pre-processing and exclusion criteria]

The dataset includes 1948 image records with AMD. Within these cases are alterations related to exudative and non-exudative AMD, such as CNV, drusen, and subretinal fluid. The remaining 2378 image records are associated with healthy patients. The images correspond to the eyes of patients with a possible AMD diagnosis, where the retina specialist has made an exhaustive clinical evaluation for the eye or eyes under study. If the patient has findings of possible AMD in both eyes, images from both eyes were included. [pre-processing and additional information]

The images were collected retrospectively from medical records between June 1st, 2019, and June 30th, 2021. Only the researchers, students, and practitioners who were involved in collecting the images had access to information that could identify the participants. In the final dataset, all records were handled anonymously. [source and additional information]

Comment 1.4 

• The training and validation process of the models lacks detail.

• Elaborate on the model training process, including data partitioning (e.g., train/test split), any cross-validation techniques used, hyperparameter tuning, and criteria for model selection. Include a flowchart or pseudocode if possible.

Author Answer Thank you for this suggestion. The flowchart of the whole process is presented in Figure 2, including the validation and model selection step.

Author Action The following paragraphs were adjusted/included in Section “The dataset and experimental setting”: 

The model evaluation was performed as follows: a nested validation strategy was made, splitting the data into a training set (70% of the samples) and a test set (30% of the samples). Ten-fold cross-validation was performed in the training set in order to define the number of epochs and to obtain an average performance. For each set of experiments given by the different combination of data sources (O

---

## [Decision Letter · Decision Letter 1]

25 Sep 2024

Interpretable Multimodal Classification for Age-related Macular Degeneration Diagnosis

PONE-D-24-15878R1

Dear Dr. Vairetti,

We’re pleased to inform you that your manuscript has been judged scientifically suitable for publication and will be formally accepted for publication once it meets all outstanding technical requirements.

Kind regards,

Xu Yanwu

Academic Editor

PLOS ONE

Additional Editor Comments (optional):

Reviewers' comments:

Reviewer's Responses to Questions

**Comments to the Author**

1. If the authors have adequately addressed your comments raised in a previous round of review and you feel that this manuscript is now acceptable for publication, you may indicate that here to bypass the “Comments to the Author” section, enter your conflict of interest statement in the “Confidential to Editor” section, and submit your "Accept" recommendation.

Reviewer #1: All comments have been addressed

Reviewer #2: All comments have been addressed

2. Is the manuscript technically sound, and do the data support the conclusions?

Reviewer #1: Yes

Reviewer #2: Yes

3. Has the statistical analysis been performed appropriately and rigorously? 

Reviewer #1: Yes

Reviewer #2: Yes

4. Have the authors made all data underlying the findings in their manuscript fully available?

Reviewer #1: Yes

Reviewer #2: No

5. Is the manuscript presented in an intelligible fashion and written in standard English?

Reviewer #1: Yes

Reviewer #2: Yes

6. Review Comments to the Author

Reviewer #1: Authors revised the manuscript based on my comments and I am satisfied with the manuscript and its in publishable form

Reviewer #2: I had commented earlier on the article and recommended that it be accepted for publication. I believe the authors have also addressed issues raised by the other reviewer. So my recommendation still remains for the article to be accepted for publication.

7. PLOS authors have the option to publish the peer review history of their article (what does this mean?). If published, this will include your full peer review and any attached files.

Reviewer #1: No

Reviewer #2: No

---

## [Editor Report · Acceptance letter]

10 Oct 2024

PONE-D-24-15878R1 

PLOS ONE

Dear Dr. Vairetti, 

I'm pleased to inform you that your manuscript has been deemed suitable for publication in PLOS ONE. Congratulations! Your manuscript is now being handed over to our production team.

Kind regards, 

on behalf of

Dr. Xu Yanwu 

Academic Editor

PLOS ONE